# The neurocomputational link between defensive cardiac states and approach-avoidance arbitration under threat
Felix H. Klaassen [1] ✉, Lycia D. de Voogd[1,2,3], Anneloes M. Hulsman[1,2], Jill X. O'Reilly[4], Floris Klumpers[1,2], Bernd Figner [1,2] & Karin Roelofs [1,2] ✉

Avoidance, a hallmark of anxiety-related psychopathology, often comes at a cost; avoiding threat may forgo the possibility of a reward. Theories predict that optimal approach-avoidance arbitration depends on threat-induced psychophysiological states, like freezing-related bradycardia. Here we used model-based fMRI analyses to investigate whether and how bradycardia states are linked to the neurocomputational underpinnings of approach-avoidance arbitration under varying reward and threat magnitudes. We show that bradycardia states are associated with increased threat-induced avoidance and more pronounced reward-threat value comparison (i.e., a stronger tendency to approach vs. avoid when expected reward outweighs threat). An amygdala-striatal-prefrontal circuit supports approach-avoidance arbitration under threat, with specific involvement of the amygdala and dorsal anterior cingulate (dACC) in integrating reward-threat value and bradycardia states. These findings highlight the role of human freezing states in value-based decision making, relevant for optimal threat coping. They point to a specific role for amygdala/dACC in state-value integration under threat.

Threat avoidance often comes at a cost, particularly in approach-avoidance conflict where avoidance may reduce the probability of aversive outcomes but also of obtaining potential rewards. Approach-avoidance conflict situations therefore require weighing of the potential reward and threat outcomes of our decisions[1–3]. Effectively arbitrating between approach and avoidance decisions is crucial for generating adaptive behavior across naturalistic environments. This process might go awry in several psychopathological conditions, including anxiety disorders as they are characterized by excessive avoidance behavior[2,4–7]. While over the past decades the decision sciences and computational psychiatry have made significant progress in modeling those value-based decisions in healthy and patient populations, the role of the threat-induced psychophysiological state of the organism has been largely overlooked[8,9].

Freezing is a defensive threat reaction characterized by immobility and heart rate deceleration (bradycardia), resulting from relative dominance of parasympathetic over sympathetic arousal in the autonomic nervous system[10,11]. There is ample evidence from postural, cardiac, and neural analyses that humans as well as animals freeze when experiencing threat[12–14].

Freezing is facilitated by projections from the central nucleus of the amygdala to the midbrain periaqueductal gray (PAG), which in turn innervates immobility and bradycardia through medullary connections to spinal cord motor neurons and the vagus nerve, respectively[15–19]. This freezing state has been shown to facilitate sensory upregulation and risk assessment while minimizing the likelihood of detection under threat[11,20–23]. For instance, freezing-related bradycardia has been linked to enhanced perceptual sensitivity[24,25] and increased action preparation[13,26,27]. Interestingly, in a recent behavioral study[28] we found evidence in line with a role of freezing states in instrumental decision-making. In this study, bradycardia was associated with value integration of reward and threat during approach-avoidance arbitration, depending on the action context. However, this relationship occurred on the subject level, and so it remains unknown how transient defensive cardiac states might affect value-based computations and underlying neural circuits on a momentary (trial-by-trial) basis. This knowledge is critical to provide starting points for optimizing interventions aiming at improved decision-making under threat in health and anxiety.

[1]Radboud University, Donders Institute for Brain, Cognition, and Behaviour, Thomas van Aquinostraat 4, 6525 GD Nijmegen, The Netherlands. [2]Radboud University, Behavioural Science Institute (BSI), Thomas van Aquinostraat 4, 6525 GD Nijmegen, The Netherlands. [3]Leiden University, Institute of Psychology and Leiden Institute for Brain and Cognition (LIBC), Rapenburg 70, 2311 EZ Leiden, The Netherlands. [4]Department of Experimental Psychology, University of Oxford, Woodstock Road, OX2 6GG Oxford, UK. ✉e-mail: felix.klaassen@donders.ru.nl; karin.roelofs@donders.ru.nl

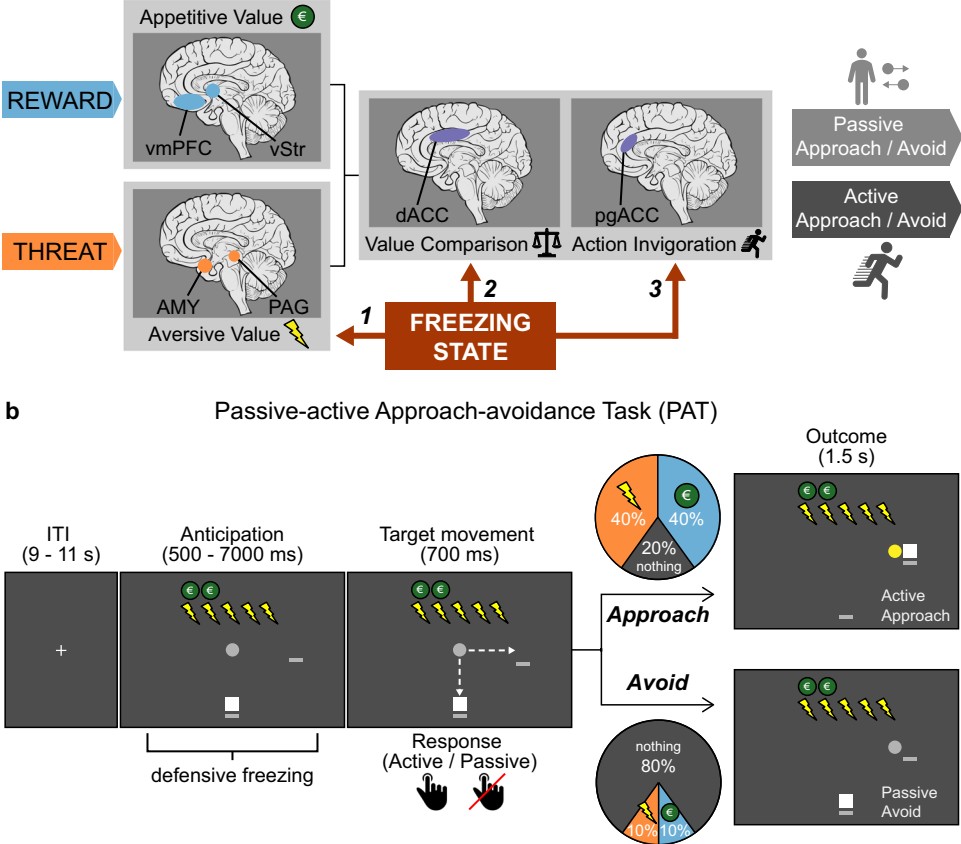

**a** Neurocomputational effects of freezing states on approach-avoidance arbitration

**b** Passive-active Approach-avoidance Task (PAT)

**Fig. 1 | Theoretical and experimental paradigm – outline of hypothesized mechanisms of freezing-state effects on approach-avoidance, and trial design of the Passive-active Approach-avoidance Task (PAT). a** Visualization of the theoretical framework in which we hypothesize three routes through which freezing states may affect the computations underlying approach-avoidance arbitration under threat; processing of aversive value in the amygdala and periaqueductal gray (AMY, PAG; route 1), value comparison in the dorsal anterior cingulate cortex (dACC; route 2), and action invigoration in the perigenual ACC (pgACC; route 3). This model hypothesizes that freezing effects on the computations in routes 1 and 2 affect the proportion of approach (vs. avoidance) choices as a function of threat (route 1) and reward-threat comparison (route 2), and the proportion of active vs. passive approach-avoidance choices (route 3). See main text for details and rationale. Note that while we here depict the ventral striatum (vStr) and AMY as encoding reward and threat information, respectively, we acknowledge that this is a simplification and that both these structures have been shown to encode threat and reward (see e.g., refs. [11,55]). Figure was adapted with permission from Livermore et al.[29]. vmPFC: ventromedial prefrontal cortex. **b** To test this model, we developed an experimental task in which participants (white square) have to approach or avoid targets (gray circle) that are associated with varying reward and threat magnitudes (ranging 1–5 euro/shocks and indicated by green coins/lightning bolts respectively). After an initial anticipation screen (duration of 6–7 s for 80% of the trials) participants could indicate their choice during the target movement window (700 ms) by positioning themselves (i.e., the white square) on the same location as where the target was moving (approach) or on the other location (avoid). If participants approached, there was a large probability to receive either the indicated number of shocks (40%) or amount of money (40%), and a small probability to receive nothing (20%). If participants avoided, there was a large probability receive to receive nothing (80%), and smaller probabilities to receive shocks (10%) or money (10%). Moreover, the target movement direction during the movement window was manipulated in two action contexts (i.e., movement towards or away from the player, 50% of all trials each), such that participants could always either actively approach/passively avoid, or passively approach/actively avoid. Participants were fully instructed about these task conditions (including the outcome probabilities). Dashed arrows were not present in the actual task.

To address this knowledge gap, we a priori formulated three potential mechanisms by which freezing states could impact approach-avoidance arbitration under threat (corresponding to routes 1–3 in Fig. 1a), previously published in Livermore et al.[29] (preregistration: https://doi.org/10.17605/OSF.IO/KYWV8). First, we hypothesize that freezing states may be associated with enhanced processing of aversive value information from potential threat in the amygdala-PAG circuit (*route 1*). The presence of environmental threat can shift value processing from being driven primarily by fronto-striatal regions (such as the ventromedial prefrontal cortex and (ventral) striatum[30–33]), to regions involved in stress coping, salience, and defensive threat reactions such as the amygdala and PAG[27,34–37], particularly during states of freezing-related bradycardia[38]. Indeed, the amygdala-PAG pathway is not only involved in triggering defensive threat reactions[13,14,16,22] but has also been implicated in processing aversive value (e.g., predicted pain[39,40]). Accordingly, we predict that stronger bradycardia states are linked to increased salience of aversive value, resulting in increased avoidance.

Second, and even more important for adaptive responding under threat, we hypothesize that freezing states may be associated with a change in how potential reward and threat are compared with each other (*route 2*). If freezing affects approach-avoidance arbitration through assessment of the associated risks[20,41], then freezing states may not be linked to the processing of aversive value alone, but rather to the value of one outcome in light of the other (e.g., discounting the value of potential reward by the value of the threat). Adapting the relative weight of potential positive vs. negative outcomes might enable more optimal decisions to approach rather than avoid when the potential reward (relative to threat) is deemed sufficiently large[28]. Such value integration has previously been suggested to be driven by the dorsal anterior cingulate cortex (dACC)[32,36,42,43] and pre-motor areas such as

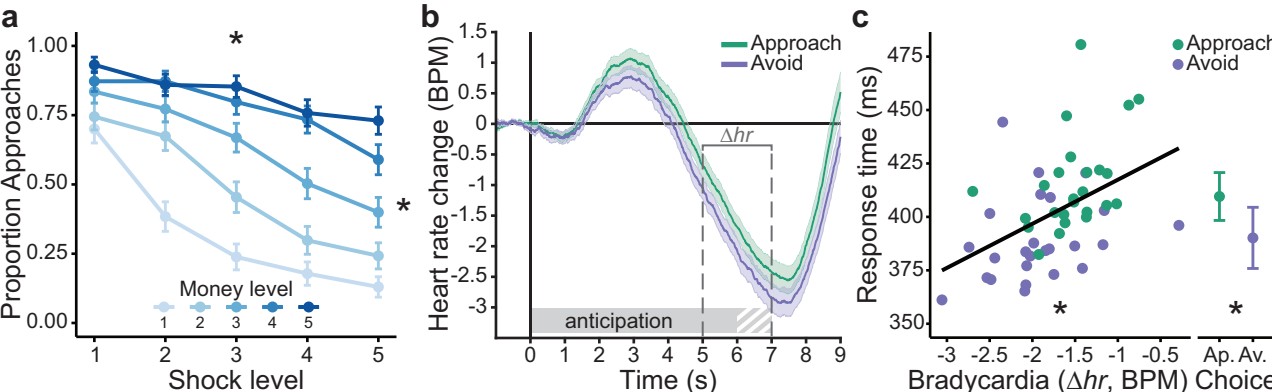

**Fig. 2 | Task effects on choice behavior and heart rate. a** Higher money and shock levels led to more approach vs avoid choices, respectively. **b** We observed a significant average heart rate deceleration during the anticipation screen relative to the 1 s pre-trial baseline indicative of a freezing-like bradycardia state, which was numerically but not significantly more pronounced in avoid (compared to approach) trials (see main text for statistics). Trial-by-trial bradycardia ($\Delta hr$) was quantified as the average baseline-corrected heart rate across a 5–7 s time window relative to the anticipation screen onset, such that lower (relative to higher) $\Delta hr$ values indicate stronger bradycardia. **c** Stronger trial-by-trial bradycardia was associated with faster response times (for illustration purposes, individual dots reflect RT and $\Delta hr$ values aggregated separately for money and shock levels, and approach and avoid choices). Moreover, we observed faster response times for avoid compared to approach decisions. Error bars indicate ±1 SEM. Gray-white striped shaded area in (**b**) reflects partial overlap between anticipation and target movement screens across different trials (i.e., movement window onset was uniformly jittered between 6–7 s relative to the anticipation screen onset). BPM: beats per minute; Ap.: approach; Av.: avoid; asterisks (*) indicate 'significant' effects (i.e., HDI$_{95\%}$ excludes 0) of money and shocks on choice (**a**) and heart rate and choice on response times (**b**).

the supplemental motor area (SMA)[29,44]. We predict that value integration in these regions might be driven by freezing states.

Third, we hypothesize that freezing states may help to prepare the body for action in acutely threatening situations through action invigoration (***route 3***). Evidence for this comes from studies linking freezing to increased fear-potentiated startle and faster response times (suggesting action preparation)[13,24,28,45]. Based on previous human work on the switch from freezing into action, we anticipate that action invigoration during freezing is associated with the perigenual region of the ACC (pgACC)[13].

To test these three routes (aversive value, value comparison, and action invigoration; Fig. 1a), 58 human participants performed a Passive-active Approach-avoidance Task (PAT)[28]. In this task people make approach-avoidance decisions in the face of varying reward (1–5 euros) and threat magnitudes (1-5 shocks) in two action contexts (passive or active; Fig. 1b). Simultaneously, we measured heart rate to assess bradycardia (as a freezing state index), and brain responses using blood-oxygen-level-dependent functional magnetic resonance imaging (BOLD fMRI). Using a modeling approach, we tested the link between event-related bradycardia states and each of the hypothesized neurocomputational mechanisms (while accounting for cardiac and respiratory noise[46]). Specifically, we modeled how approach-avoidance decisions varied as function of bradycardia interactions with the number of shocks (aversive value), the money-shock level difference (value comparison), and the action context (action invigoration)[29]. Neurally, we expected bradycardia interactions with aversive value to activate the amygdala and PAG, with value comparison to engage the dACC, and with action invigoration to involve the pgACC (Fig. 1a).

## Results

### Bradycardia states are associated with stronger shock-induced avoidance

**Approach-avoidance decisions under threat.** In the PAT, participants were well able to trade off shocks versus money, replicating typical choice and response time patterns[28] (Fig. 2a, see **Methods** Eqs. (1) and (2) for Bayesian mixed-effects model, i.e. BMM, specifications).

Regarding choice, higher money and shock levels led to more approach and avoidance, respectively (as preregistered, statistical effects whose Bayesian 95% and 90% posterior highest density interval 'HDI' did not include 0 were interpreted as 'significant' and 'marginally significant',

respectively, see Methods for details; $B_{money} = 1.54$, HDI$_{95\%}$ = [1.26, 1.83]; $B_{shocks} = -1.10$, HDI$_{95\%}$ = [−1.35, −0.84]; Fig. 2a). Additionally, shock-induced avoidance was more pronounced for lower money levels, and money-induced approach was less pronounced for lower shock levels ($B_{money:shocks} = 0.40$, HDI$_{95\%}$ = [0.22, 0.57]). Choice was not affected by the action context ($B_{actioncontext} = 0.06$, HDI$_{90\%}$ = [−0.04, 0.17]), but the action context did interact with the money and shock levels. Specifically, the approach-effect of money was stronger in active compared to passive action contexts, indicating action invigoration as a function of reward ($B_{money:actioncontext} = -0.27$, HDI$_{95\%}$ = [−0.40, −0.16]). Conversely, the avoidance effect of shocks was stronger in passive compared to active action contexts, indicating more passive responses (i.e., action inhibition) as a function of threat (though only marginally so; $B_{shocks:actioncontext} = 0.09$, HDI$_{90\%}$ = [0.008, 0.17]).

Differences between approach vs. avoidance decisions were also reflected in the response times. Participants responded faster for avoid compared to approach choices ($B_{choice} = -0.04$, HDI$_{95\%}$ = [−0.07, −0.02]; Fig. 2c), and this response time difference depended on the reward and threat magnitudes: Higher money levels led to faster approach compared to avoid responses ($B_{money:choice} = 0.04$, HDI$_{95\%}$ = [0.02, 0.05]), while higher shock levels led to slower approach compared to avoid responses ($B_{shocks:choice} = -0.04$, HDI$_{95\%}$ = [−0.06, −0.02]).

In sum, we observed (as previously[28]) a balanced trade-off between reward and threat, and interactions of reward and threat magnitudes with action and response times.

**Behavioral interactions with bradycardia states.** Next, we assessed to what extent our experimental paradigm induced the desired cardiac effects, and whether these heart rate dynamics interacted with participants' task behavior.

A significant reduction in heart rate during the anticipation screen relative to the pre-trial baseline indicated the expected freezing-like bradycardia state ($B_{intercept} = -1.77$, HDI$_{95\%}$ = [−2.12, −1.42]). This bradycardia response was not affected by varying money levels ($B_{money} = 0.10$, HDI$_{90\%}$ = [−0.004, 0.20]), shock levels ($B_{shocks} = -0.02$, HDI$_{90\%}$ = [−0.12, 0.07]), the action context ($B_{actioncontext} = 0.04$, HDI$_{90\%}$ = [−0.06, 0.15]), nor their interactions (BMM on $\Delta hr$, Methods Eq. (3); Fig. 2b).

Moreover, stronger bradycardia *was* related to an increased effect of shock level on avoidance ($B_{shocks:heartrate} = 0.08$, HDI$_{95\%}$ = [0.001, 0.17]).

**Fig. 3 | Neural correlates of reward-threat and approach-avoidance anticipation. a** We observed positive correlations between BOLD and money levels in the ventral striatum (vStr), and between BOLD and shock levels in the supplemental motor area/dorsal anterior cingulate (SMA/dACC) and anterior insula. **b** We observed higher BOLD activity in the ventral striatum, amygdala, and ventromedial prefrontal cortex (vmPFC) for approach compared to avoid choices. Images are thresholded at $p < .001$ whole-brain uncorrected for display purposes. All labeled areas are significant at $p < .05$ FWE-corrected.

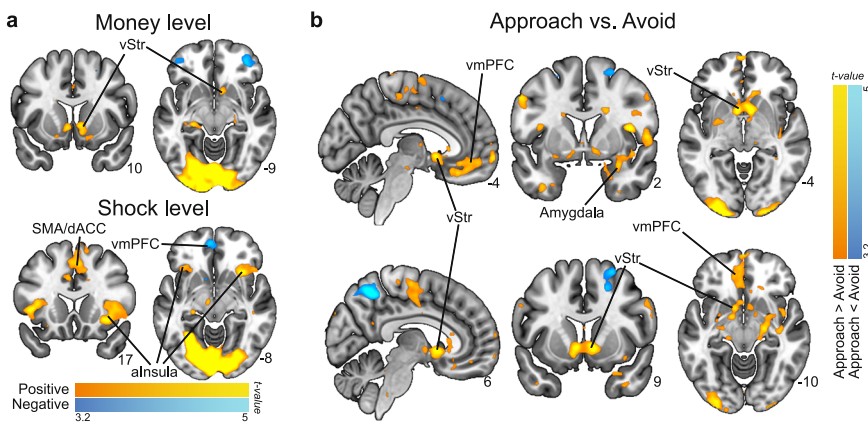

Specifically, in trials with more bradycardia, more shocks led to increasingly more avoidance. There were no significant interactions of bradycardia with any other experimental conditions, including money or the money-by-shocks interaction.

Finally, stronger bradycardia was associated with faster response times (BMM on RTs, Methods Eq. (5); $B_{heartrate} = 0.03$, $HDI_{95\%} = [0.007, 0.05]$; Fig. 2c).

Together, these results show that bradycardia states were related not only to general task involvement, but also approach-avoidance decisions and faster responding.

### Distinct neural networks underlying anticipation of reward and threat magnitudes versus approach-avoidance decisions

**Tracking of reward and threat magnitudes during anticipation.** Before detailing the modeling effects, we investigated the task-based neural correlates of reward and threat anticipation, by performing a parametric modulation of the fMRI signal as a function of the money and shock levels (Fig. 3a). For reward, as expected, higher money levels were positively correlated with increased BOLD activity in the ventral striatum (left: $p = 0.003$, right: $p < 0.001$; peak-voxel family-wise error small-volume corrected, i.e., FWE-SVC). We also expected a reward effect in the vmPFC, but we did not observe a significant association ($p = 0.059$ peak-voxel FWE-SVC within an anatomical vmPFC mask, see Methods). Money levels were negatively correlated with BOLD in the right lateral orbitofrontal cortex ($p = 0.024$ cluster-level FWE-corrected). Regarding threat, shock levels were as expected positively correlated with BOLD activity in regions part of the salience network such as SMA/dACC and anterior insula, and regions part of the executive control network such as the dlPFC and precuneus (all $p < 0.001$ cluster-level FWE-corrected). This contrast did not show the expected effects of the shock levels in subcortical regions such as the amygdala; this was only the case when contrasting approach vs. avoid trials, i.e. when the probability of receiving a shock is relatively high (see below). Finally, shocks were negatively correlated with BOLD in the vmPFC ($p < 0.001$ cluster-level FWE-corrected; for all statistics, see Supplementary Table 1). Together, these results show that cortico-striatal regions traditionally implicated in reward and threat confrontation (e.g., reception of money vs. shocks) also parametrically track the processing of these potential outcomes during anticipation.

**Circuits underlying approach-avoidance decisions.** Approach (compared to avoid) trials showed increased activation in regions implicated in both appetitive and aversive processing, such as the ventral striatum (left and right: $p < 0.001$ peak-voxel FWE-SVC) as well as the amygdala (left: $p = 0.005$, right: $p < 0.001$; peak-voxel FWE-SVC). Additionally, regions previously implicated in (approach-avoidance) decision-making such as the vmPFC and the right hippocampus showed

stronger BOLD responses during approach (both $p < 0.001$ cluster-level FWE-corrected). The inverse contrast—showing stronger activation for avoid vs. approach trials—revealed activity in regions consistent with the frontal-parietal network: the right precuneus and superior frontal gyrus (precuneus: $p < 0.001$, sup. front. gyr.: $p = 0.007$; cluster-level FWE-corrected; Fig. 3b; see Supplementary Table 2 for all statistics). Finally, neural patterns were similar for active vs. passive approach-avoidance decisions (see Supplementary Table 2).

To verify relevance of these neural findings for decision-making, we performed a follow-up hemodynamic response function (HRF)-agnostic time-series analysis of BOLD response patterns (i.e., a finite impulse impulse response analysis). This control analysis indicated that the neural approach-avoidance effects in our main regions (ventral striatum, amygdala, and vmPFC) were only related to the choice and not the subsequent response (see Supplementary Fig. 1).

Collectively, our task-based neural results outline an approach-avoidance circuit consistent with previous work, and show distinct networks underlying approach-avoidance arbitration and anticipatory processing of reward/threat magnitudes, with only partial overlap.

### Neural circuits underlying the link between bradycardia states and approach-avoidance decisions

**Support for aversive value and value comparison models.** After having verified the task-based effects, we addressed our main experimental question regarding the link between bradycardia states and the computations underlying approach-avoidance arbitration (i.e., aversive value, value comparison, and action invigoration, Fig. 1a), using computational modeling and model-based fMRI analysis.

We created three separate models (further referred to as freezing models) capturing the effects of aversive value (AV), value comparison (VC) and action invigoration (AI) on trial-by-trial approach-avoidance choices, and contrasted them against a base model (following previous work[47–49]). The AV model included an interaction term between bradycardia and the number of shocks (i.e., $\beta_{s:hr}$). The VC model included an interaction between bradycardia and money-shock difference $\Delta ms$, which we mathematically defined as the difference between the money and shock levels on offer (i.e., $\beta_{ms:hr}$, and $\Delta ms = money - shocks$). Lastly, the AI model featured an interaction term between bradycardia and the action context, which estimates participants' probability of giving an active (vs. passive) response (i.e., $\beta_{ac:hr}$). Details on the modeling specifications are included in the Methods.

We assessed whether the freezing models captured the observed patterns in the choice data (Fig. 4, upper panels a–c show aggregated raw choice proportions, lower panels d–f show model-predicted choice probabilities which were generated from model-estimated decision values 'DVs', where DVs < 0 correspond to $p$(approach) < 0.5, and DVs > 0 correspond to $p$(approach) > 0.5; note that low vs. high $\Delta hr$ values indicate trials with

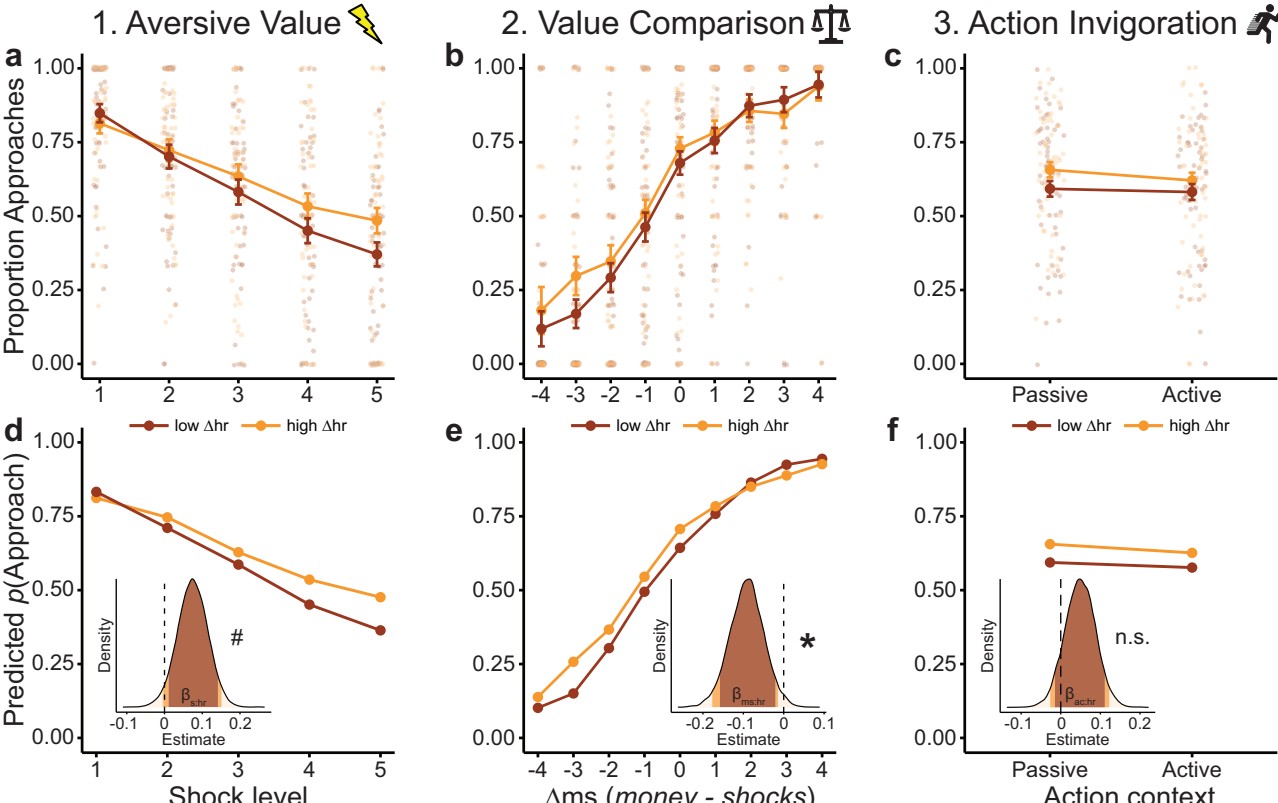

**Fig. 4 | Comparison of observed and model-predicted bradycardia-state interactions underlying approach-avoidance choices.** Observed (**a**–**c**) and model-predicted (**d**–**f**) effects on choice for the three hypothesized routes (1: aversive value, 2: value comparison; 3: action invigoration). Specifically, we plot the interactions between bradycardia and the number of shocks (**a**, **d**), the money-shock difference $\Delta ms$ (**b**, **e**), and the action context (**c**, **f**). Model-based plots also display the posterior distributions of the interaction coefficients, with the 90% HDI shaded in red, and the 95% HDI in orange. For plotting purposes only, we created conditions with stronger

bradycardia (low $\Delta hr$) and weaker bradycardia (high $\Delta hr$; respectively containing the trials with the 33% lowest vs highest $\Delta hr$ values). Model predictions are plotted as predicted approach probabilities. In (**a**–**c**), small, semi-transparent dots represent individual participant data ($n = 58$ per condition). Error bars indicate ±1 SEM; asterisks (*) indicate 'significant' effects (i.e., $HDI_{95\%}$ excludes 0), hash icons (#) indicate 'marginally significant' effects (i.e., $HDI_{90\%}$ excludes 0), n.s.: not 'significant' (i.e., $HDI_{90\%}$ includes 0); m: money; s: shocks.

stronger vs. weaker bradycardia, respectively). Note that some of the bradycardia effects in the freezing models (i.e., $\beta_{s:hr}$ and $\beta_{ac:hr}$) were also tested above in a single task-effect BMM. Here, these interactions are entered in separate models to allow us to isolate their unique contribution to the neural data (see below).

First, the aversive value model confirmed the relationship between bradycardia and shocks (also tested in the BMM above; $\beta_{s:hr} = 0.07$, $HDI_{90\%} = [0.01, 0.14]$, Fig. 4a). For the value comparison model, we now found a significant interaction between bradycardia and $\Delta ms$ ($\beta_{ms:hr} = -0.09$, $HDI_{95\%} = [-0.18, -0.01]$), revealing a more pronounced effect of money-shock level differences on choice in trials with stronger compared to weaker bradycardia (Fig. 4b). Interestingly, this interaction seems particularly driven by a higher approach rate in trials with high money (relative to shock) levels (i.e., high $\Delta ms$ values), and strong bradycardia. In all other conditions (i.e., $\Delta ms$ values below 2), bradycardia was mostly associated with more avoidance (regardless of the $\Delta ms$ value). On trials where expected reward strongly outweighs the threat magnitude, bradycardia was associated with increased, rather than decreased approach. Whereas on trials with comparable reward-threat magnitudes, or trials where the threat magnitude outweighs the expected reward, stronger bradycardia is associated with more avoidance. Finally, in the action invigoration model there was no relationship between bradycardia and the action context (also tested in the BMM above; $\beta_{ac:hr} = 0.05$; $HDI_{90\%} = [-0.01, 0.11]$, Fig. 4c). Overall, the freezing models captured the observed behavioral patterns well (quantitative model comparison indicated that all models fit the data roughly equally well, with the aversive value

model slightly outperforming the other models, including the base model; see Supplementary Table 3).

**Model-based and task-based fMRI analyses reveal highly similar neural circuits.** Then, for our model-based fMRI analysis approach, we first investigated to what extent the predicted approach-avoidance choices by the base model generated similar neural circuits of BOLD activation as the observed approach-avoidance choices (Fig. 5a). Here we found indeed a highly similar pattern of regions for the model-based analysis: a positive correlation between higher decision values (i.e., higher predicted probability to approach) and BOLD activity in the ventral striatum (left: $p = 0.001$, right: $p = 0.009$ peak-voxel FWE-SVC), right amygdala ($p = 0.019$ peak-voxel FWE-SVC), and vmPFC ($p < 0.001$, cluster-level FWE), and a negative correlation (i.e., higher predicted probability to avoid) with BOLD in the right precuneus and superior frontal gyrus (precuneus: $p = 0.001$, sup. front.: $p = 0.008$, cluster-level FWE; see Supplementary Table 4 for all significant clusters). This supports the base model as being a solid foundation to investigate the neural correlates of the subsequent freezing models each incorporating the distinct bradycardia-state mechanisms.

**Amygdala and dACC involvement in bradycardia-state interactions with aversive value and value comparison.** Next, to test our core neural hypotheses, we investigated the unique contribution of the freezing models to the observed BOLD-fMRI signal, as compared to the base model (Fig. 5). For each of the three freezing models, we computed a

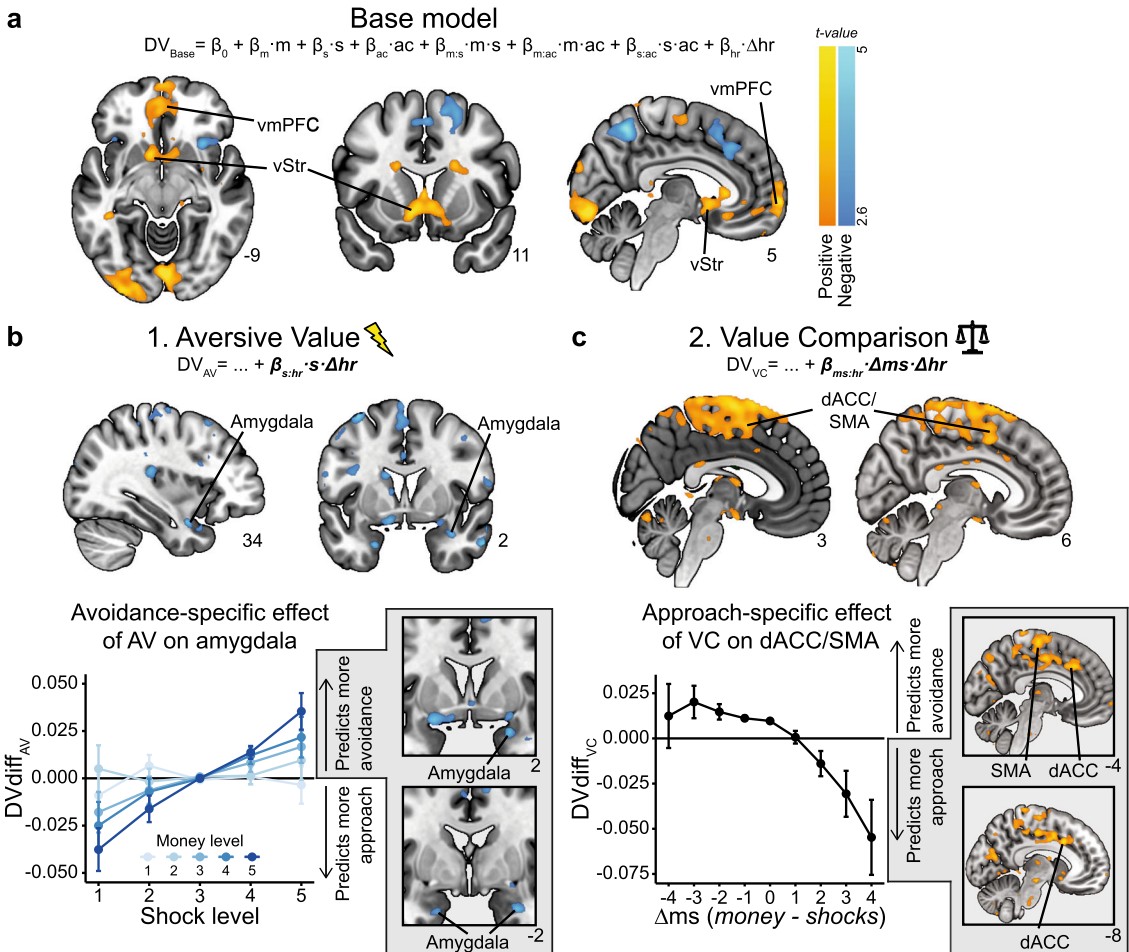

**Fig. 5 | Neural circuits underlying the link between bradycardia states and value-based computations. a** For the base model, we find positive correlations between model-extracted decision values (DVs, indicating the value of approaching vs. avoiding) and BOLD activity in the ventral striatum (vStr) and ventromedial prefrontal cortex (vmPFC), delineating a similar neural circuit as was observed from the task-based analysis (see Fig. 3). **b** DVs estimated from the aversive value (AV) model were, relative to the base model, negatively correlated with BOLD activity in the amygdala (top). A follow-up analysis revealed that this negative relationship was specific to conditions in which AV predicted more avoidance, compared to the base model (i.e., DVdiff_AV > 0; bottom). **c** DVs estimated from the value comparison

(VC) model were, relative to the base model, positively correlated to BOLD in a large whole-brain significant cluster in the dorsomedial prefrontal cortex (dmPFC), spanning the SMA and dACC (top). Follow-up analysis revealed that this relationship was specific to conditions in which VC predicted more approach, compared to the base model (i.e., DVdiff_VC < 0; bottom). DVdiff_AV/VC: trial-wise difference scores of the DVs of AV/VC models relative to the base model. Positive/negative DVdiff_AV/VC values indicate stronger predicted probability to avoid/approach by AV/VC models compared to the base model. Error bars indicate ±1 SEM. Images are thresholded at $p < 0.005$ whole-brain uncorrected for display purposes. All labeled areas are significant at $p < 0.05$ FWE-corrected.

regressor containing difference scores that quantify to what extent that model predicts different decision values as compared to the base model (i.e., $DVdiff_{freezemod} = DV_{Base} - DV_{freezemod}$). For a given freezing model, positive difference scores (i.e., $DVdiff_{freezemod} > 0$) indicate that it predicts a higher probability of avoidance in that trial, and negative difference scores (i.e., $DVdiff_{freezemod} < 0$) indicate it predicts a higher probability of an approach choice (relative to the base model, see Methods). Note that these prediction differences between the freezing models and the base model are driven by the distinct bradycardia-state interaction terms, since all models were otherwise parameterized identically. The $DV_{Base}$ and three $DVdiff_{freezemod}$ vectors were then together entered as parametric modulators of the BOLD signal during the anticipation screen. All parametric modulators were orthogonalized relative to the unmodulated (i.e., anticipation screen) regressor, such that each regressor captures its unique BOLD-signal variance (i.e., shared variance between two regressors will be attributed to the error term[50]).

For aversive value (AV), we observed a negative correlation between its difference scores and BOLD activity in the amygdala (as expected; right: $p = 0.022$ peak-voxel FWE-SVC; survives Bonferroni correction for two ROIs, i.e., $p_{corrected} = 0.044$), but unexpectedly not in the PAG (Fig. 5b; for

whole-brain results see Supplementary Table 4). This suggested that as the difference in choice predictions between the AV and base model increased (i.e., as $DVdiff_{AV}$ becomes more positive), BOLD activity in the amygdala decreased. However, it remained unclear whether this was a general relationship (along the whole range of positive and negative trial-wise $DVdiff_{AV}$ values), or whether this relationship was specific to certain trial conditions. To further investigate this, we performed a follow-up analysis in which we split the AV difference regressor in two: one for trials in which the model predicted a higher probability of avoidance compared to the base model ($DVdiff_{AV} > 0$), and one for trials in which it predicted a higher probability of an approach choice compared to the base model ($DVdiff_{AV} < 0$). This revealed that only in the trials in which AV predicted more avoidance compared to the base model (i.e., $DVdiff_{AV} > 0$), there was a negative relationship (i.e., stronger prediction of avoidance) with BOLD in the amygdala (right: $p = 0.014$, $p = 0.018$, $p = 0.027$, left: $p = 0.045$; all peak-voxel FWE-SVC; Fig. 5b). This suggests that the relative decrease of amygdala activity is specific to the interaction effect of shocks and bradycardia on avoidance captured by the AV model (Fig. 4a).

For value comparison (VC), we observed a large positive whole-brain significant cluster in the dorsomedial prefrontal cortex (dmPFC) spanning

the (pre) supplemental motor area (SMA) and—as we predicted—the dACC ($p < 0.001$ cluster-level FWE; Fig. 5c). Follow-up analysis revealed that this effect was only significant in conditions in which the VC model predicted a higher probability to approach compared to the base model, which corresponds to trials with $\Delta ms \geq 2$ ($p < 0.001$ peak-voxel FWE for dACC ROI, $p = 0.004$ cluster-level FWE-corrected for whole brain; Fig. 5c; see Supplementary Table 4 for all statistics). Together, these results suggest that the bradycardia-related increase in approach choices for highly positive money-shock differences (Fig. 4b) is inversely related to BOLD activity in the dACC/SMA area (note that more negative $DVdiff_{VC}$ values indicate higher differential predicted approach probabilities, see Fig. 5c). These findings might point to a reduction in conflict-related processing when the money-shock difference is relatively high, since approach-avoidance conflict is highest when money and shock levels are similar, and lowest when they are maximally dissimilar (e.g., the average approach rate is 48% when $\Delta ms = -1$, and 13% / 93% when $\Delta ms = -4/4$). Finally, for action invigoration (AI), we did not observe any significant associations in our region of interest (i.e., pgACC), nor across the rest of the brain.

## Discussion

This study highlights the role of the psychophysiological state of freezing in approach-avoidance arbitration under threat, establishing a neurocomputational link between bradycardia states and approach-avoidance decisions. Participants made passive and active approach-avoidance decisions in the face of varying reward and threat magnitudes, showing increased approach for higher money levels and increased avoidance for higher shock levels. Moreover, bradycardia states during anticipation indicated that participants were in a psychophysiological state of freezing. Neurally, approach-avoidance arbitration was associated with BOLD-response patterns in the amygdala, ventral striatum, and vmPFC (in line with previous reports[30,32,36,42,51,52]).

Additionally, we found that trial-by-trial bradycardia states were linked to value-based computations underlying approach-avoidance arbitration. First, bradycardia states were associated with a stronger effect of threat of shock (i.e., aversive value) on avoidance decisions, which was negatively associated with BOLD activity in amygdala. Secondly, bradycardia states were linked to a stronger tendency to approach rather than avoid when the expected reward outweighs the threat (i.e., value comparison), an effect associated with the dACC and SMA. Together, we delineate a neural circuit involved in approach-avoidance arbitration under threat, with specific involvement of the amygdala and dACC in integrating subjective outcome values and defensive psychophysiological states during approach-avoidance decisions. These results stress the relevance of the organism's psychophysiological state in approach-avoidance arbitration under threat, fitting with recent empirical and theoretical work[11,24,28,29]. Together, we highlight the role of human freezing states in value-based decision processes relevant for optimal threat coping.

Bradycardia states are linked to distinct value-based computations underlying approach-avoidance arbitration, with distinct neural signatures. Specifically, using model-based fMRI analysis, we found that the relation of bradycardia states with aversive value involves the amygdala, and that the link with value comparison (i.e., reward-threat magnitude differences) involves the dACC/SMA.

First, the finding that bradycardia was linked to increased shock-induced avoidance, indicates that on trials with stronger bradycardia higher shock levels led to more avoidance compared to trials with less bradycardia. This finding points to increased sensitivity for aversive value during bradycardia states, resulting in a stronger tendency to avoid. Although the relation with BOLD activity in the amygdala was expected, the direction of the correlation was negative, rather than positive. Perhaps amygdala deactivation during bradycardia states might signal an increase in attentional processing under threat, which is then accompanied by increased sensitivity to aversive value. Previously, downregulation of the default mode network (DMN) during cognitively demanding tasks has been shown to be accompanied by amygdala inhibition[53,54]. Such amygdala deactivation has

also been observed in conditions under threat[55,56]. Similarly, such an increase in attentional processing under threat might explain why sensory upregulation occurs during bradycardia states[24,38]. Alternatively, the negative relationship with amygdala activity might reflect a decrease in outcome-related risk anticipation, a safety signal, or sense of relief after choosing to avoid[57]. Indeed, a potential role of the amygdala in value-based decision-making might be to process the potential negative consequences of the decision, such as anticipation of risk and potential negative outcomes[41,58–60]. As such, the reduction in amygdala activity might reflect a consequence of the decision (to avoid) rather than a mechanism underlying the decision itself. Unfortunately, the limited temporal specificity of the fMRI signal does not allow us to separate these two possible explanations. However, our FIR time-series analysis of the BOLD response in the amygdala showed that choice effects already occurred early during the anticipation window (see Supplementary Fig. 1), hinting at early involvement of the amygdala in the approach-avoidance decision process. Future studies using neuroimaging methods with higher temporal precision, such as magnetoencephalography (MEG)[61–64], might provide more insight into the role of the amygdala and associated neural circuits in value-based decision-making under threat. Overall, these findings provide further support for an important role of the amygdala in approach-avoidance arbitration in rodents and humans[65,66], but also reveal that threat states may underlie this decision-making process.

Secondly, bradycardia states were linked to value comparison of potential reward and threat, reflected by a more pronounced impact of the reward-threat magnitude difference on choice. Namely, trial-by-trial bradycardia was associated with more avoidance when the expected reward was equal to or less than the threat, while bradycardia was linked to more approach when the expected reward outweighed the threat. This behavioral observation is in support of the notion that bradycardia states are not only related to value processing in the aversive domain (i.e., defensive responses) but also the appetitive domain (i.e., approach behavior[67]). Whereas previously such association between computations of subjective outcome value and bradycardia states was found across participants[28], we now show this on a trial-by-trial level. We thus extend previous work[28], illustrating how the current physiological state of the decision maker may inform approach-avoidance arbitration. This finding provides empirical support for recent theoretical work proposing that the presumed role of the psychophysiological state of freezing in optimizing decision-making under threat[11,29], namely to approach when the expected reward outweighs threat, so as not to miss out on potential opportunities.

The relationship between bradycardia states and value comparison was associated with BOLD-activity changes in the dACC/SMA. Specifically, trials with stronger bradycardia and with high reward (relative to threat) magnitudes were associated with less dACC/SMA BOLD activity. This might reflect a change in conflict-related processing. This interpretation would be in line with other evidence indicating a role for the dACC in conflict resolution[30,32,42,68]. Indeed, in our paradigm conflict is relatively high when the reward and threat magnitudes are similar, while conflict is relatively low when the expected reward outweighs threat. Additionally, recent work has demonstrated a central role of the dmPFC (specifically the dACC and preSMA) in encoding value-based decision variables, such as the expected value[43,44]. The fact that we find that similar regions are involved in the interaction between value comparison and bradycardia states, suggests that threat states are linked to the neurocomputational basis of value-based decisions in general, and not just approach-avoidance decisions specifically. Finally, while previous work has linked parasympathetic activity to the vmPFC and sympathetic activity to the dACC[69], we now show that parasympathetic effects (i.e., bradycardia) might also affect processes in other more dorsal prefrontal areas, such as the dACC/SMA. Together, these findings illustrate the frontal cortical circuitry supporting bradycardia-state interactions with the value computations underlying approach-avoidance arbitration.

Finally, we found no evidence for a relation between bradycardia states and action invigoration in terms of active versus passive decision-making. There was no behavioral interaction between bradycardia and action

context, and accordingly there were no significant correlates at the neural level. This means that, within our paradigm, bradycardia states might not play a role in the decision to take an action. However, we did find a trial-by-trial relationship between bradycardia and faster response times, replicating previous work[23,26–28,70]. This might suggest that bradycardia states only play a role in actions through increased speed of action once the organism has decided to act.

One open question pertains to the causal chain of effects regarding the link between freezing states and value-based computations underlying approach–avoidance. Indeed, the interpretation of our findings is limited by the correlational nature of our experimental design. For example, freezing states might not causally contribute to changes in value computations but rather be an epiphenomenon of task engagement. If that were the case, one would expect bradycardia to be correlated with general task effects, such as arousal and effort. However, while threat and reward typically both induce arousal (e.g., through increased skin conductance responses), bradycardia is particularly observed as a function of threat and not reward magnitudes[28,71,72], ruling out a general effect of arousal. Additionally, bradycardia states are commonly associated with faster response times, which is typically indicative of easier rather than more effortful decisions[13,28,73,74]. How should we then interpret the relation between bradycardia states and value computations? We speculate that freezing states may be more than an epiphenomenon, and that they serve a role in integrating the value of external stimuli while accounting for the internal bodily state. This interpretation would be in line with findings that heart rate reductions and slow breathing rates during freezing serve sensory processing, for example by optimizing interoception of cardiac and other bodily signals[75], subsequently increasing neural signal-to-noise[24,76,77]. This sensory upregulation might subsequently facilitate neural processing in downstream brain circuits involved in value assignment. This way, we theorize, freezing states may facilitate the computation of aversive value in the amygdala (increasing threat avoidance) as well as value comparison of reward and threat in the dmPFC (maintaining the behavioral flexibility to obtain large rewards). Indeed, our findings imply that how we weigh the potential outcomes of our actions depends on the current psychophysiological state. Nevertheless, causal manipulations, such as deep brain stimulation of regions critically involved in freezing states (like the periaqueductal gray[16,18,78]), are needed to directly test this hypothesis.

Our findings may have implications for people suffering from internalizing symptoms, including anxiety and depression, who display excessive and costly avoidance[7,51]. Indeed, altered freezing has been associated with psychopathology[79–81]. For example, infants who displayed little to no freezing at all early in life were at high risk to develop internalizing symptoms up until late adolescence[82,83]. We provide a formal framework that could be used to test transdiagnostically what might drive maladaptive avoidance patterns[6,8,84]. For example, it would be important to test whether avoidance in anxiety and depression are related to altered freezing patterns and if this is driven by changes in the computation of threat value, and reward-threat comparison.

In conclusion, this work illustrates how the psychophysiological state of bradycardia is linked to the neurocomputational underpinnings of approach-avoidance arbitration under threat. Our findings suggest that during bradycardia states the amygdala is involved in increased sensitivity to aversive value (reflected by increased shock-induced avoidance), while the dACC/SMA is involved in the comparison of appetitive vs. aversive value. The results thus demonstrate a potential role for psychophysiological states in shaping value-based decision-making under threat. Altogether, our findings support recent theorizing on an active role of human freezing states in optimizing decision-making under threat[11], and may contribute to a better understanding of how defensive threat reactions may affect cognitive and affective processes.

## Methods
This study was preregistered before data analysis on the Open Science Framework (https://doi.org/10.17605/OSF.IO/KYWV8). All research

activities were carried out in accordance with the Declaration of Helsinki, approved by the local ethics committee (Ethical Reviewing Board CMO/METC [Institutional Research Review Board] Arnhem-Nijmegen, CMO 2014/288), and all ethical regulations relevant to human research participants were followed.

### Participants
Sixty-seven healthy volunteers completed the study. After data collection, 9 participants were excluded from data analysis due to an imaging artifact ($n = 3$), too little variance in choice behavior ($n = 3$), unusable heart rate data ($n = 2$), or falling asleep during the experiment ($n = 1$), leading to a sample size of $n = 58$ participants (aged 18–34, [M ± SD = 24.17 ± 3.43], 41 females). Inclusion criteria were age (between 16 and 35 years), Dutch or English speaking, and right-handedness; exclusion criteria were MR incompatibility, self-reported current pregnancy, current or lifetime history of psychiatric, neurological, or cardiovascular disorder, endocrine illness or treatment, claustrophobia, epilepsy, and self-reported high or low blood pressure. All participants gave written informed consent before participation, and were paid for participation (€24) plus bonus money contingent on their task choices (max €15, see PAT section below).

### Experimental design and procedure
Participants came to the lab for a single session of 150 min. Upon arrival, they read and signed the screening and informed consent forms. Participants then read through onscreen task instructions and performed 8 practice trials of the Passive-active Approach-avoidance Task (PAT; see below) in a behavioral cubicle. All participants performed the same set of practice trials. Afterwards, the researcher made sure participants fully understood the task by asking them to verbally report (and, if necessary, correcting) their understanding of how to approach/avoid, and the relation between approach-avoidance and the probability of receiving one of the outcomes. Next, participants were escorted to the MRI lab, attached to the measurement and stimulation electrodes, given the shock-workup procedure (see Electrical stimulation section below), and placed inside the scanner. In the scanner, we acquired BOLD fMRI images while participants performed the PAT in three runs of 62 trials each (186 in total). Each run consisted of one repetition of all money (5), shock (5), and action context (2) combinations with long anticipation screen durations (i.e., ≥6 s; 50 trials in total), and 12 trials with short anticipation screen durations (pseudorandomized money/shock/action context parameters). See PAT section below for details. A field map was acquired between run 1 and 2, and an anatomical image was acquired after the 3rd run. After the last scan, participants had to subjectively rate the intensity of one final electrical shock (ranging 1 [not painful at all] to 5 [very painful]), after which they filled in three questionnaires on a computer outside of the scanner: the Beck Depression Inventory (M ± SD = 6.08 ± 5.98[85]) and the Trait questionnaire of the STAI (M ± SD = 37.27 ± 10.17[86]) to characterize our sample, as well as a short debriefing form.

### Passive-active approach–avoidance task (PAT)
Participants performed the PAT[28]. In this computer task, participants were instructed to make approach–avoidance decisions in response to a moving target that was associated with receiving varying amounts of monetary rewards (1–5 euro) and numbers of shocks (1-5 stimulations). Additionally, depending on the action context, they could approach or avoid either actively or passively (Fig. 1B).

In each trial, after an intertrial interval of 9–11 s, the participant was first shown the anticipation screen detailing the money and shock levels (ranging 1–5 each, with 5*5 = 25 possible combinations), the target (a gray circle in the center), and the player icon (a white square in the bottom). This information remained onscreen for a variable anticipation-to-movement screen interval (AMI) of 500–7000 ms. Because of the slow temporal development of the heart rate signal, we only included trials with long (i.e., 6000–7000 ms) AMIs in our analyses (80% of trials per participant). The short (i.e., 500–5500 ms) AMI trials were in the PAT to ensure continued

activation of the participant[13,28,70]. After the AMI, the target started to move either towards (i.e., downwards) or away from the player icon (i.e., left or right, pseudorandomized across trials). During the target movement window (700 ms), the participant could either approach the target by positioning the player icon at the same location as the target, or avoid the target by positioning the player icon at the other location. Specifically, if the target moved downwards (passive action context) the participant could either passively approach (by withholding a response) or actively avoid (by pressing a button). If the target moved left/right (active action context), the participant could either actively approach (by pressing a button) or passively avoid (by withholding). Depending on the approach/avoidance choice, this led probabilistically to either money, shocks, or no outcome. Specifically, if the participant approached, there was a high probability of receiving either the specified money amount (40%) or number of shocks (40%), and a low probability of no outcome (20%). Inversely, if the participant avoided, there was a high probability of receiving no outcome (80%) and a low probability of receiving the money (10%) or shocks (10%). The selected outcome was presented during the outcome screen (1.5 s) by color coding of the target stimulus (i.e., green for money, yellow for shocks, gray for no outcome), and shocks were paid out immediately during that color change. The summed monetary outcome of three randomly selected trials (max. €15) was paid out as a bonus fee. The task was programmed in MATLAB[87] using the Psychtoolbox extension[88].

### Psychophysiological measurements

To measure heart rate, a pulse oximeter was affixed to the first (i.e., index) finger of the left hand. To measure skin conductance, two standard Ag/AgCl electrodes were applied to the distal phalanges of the third and fourth (i.e., ring and pinky) fingers of the left hand. Respiration was measured using a respiration belt placed around the participant's abdomen. These were all recorded through a BrainAmp MR amplifier at a sampling rate of 1000 Hz in BrainVision Recorder (Brain Products GmbH). An EyeLink 1000 eyetracker system (SR Research, Kanata, Ontario, Canada) was used to measure pupil diameter (sampling rate of 1000 Hz). Skin conductance and pupil/eye data were not analyzed for this study because of relatively poor data quality for several participants.

### Electrical stimulation

Electrical shocks were delivered through two 10 mm Ag/AgCl electrodes attached to the distal phalanges of the third and fourth fingers of the right hand using a MAXTENS 2000 (Bio-Protech) machine. Shock durations were 200 ms (consisting of a train of 250 μs pulses at 150 Hz), delivered at an intensity ranging from 0–40 V/0–80 mA divided in pre-defined 10 steps/ levels. To determine the appropriate stimulation intensity per participant, we performed a standardized shock calibration procedure (see refs. 28,89). Each participant received and subjectively rated exactly five shocks (always starting at intensity level 2), with the aim to converge at a shock intensity level that was experienced as uncomfortable, but not painful (i.e., rated as 4 on a scale ranging from '1 = not painful at all', to '5 = very painful'). In brief, if a shock was rated as 3 or lower the shock intensity was increased, if a shock was rated as 5 the intensity was decreased, and if it was rated at 4 the intensity was kept the same. The average final shock intensity across shock subjects was 4.48 ± 1.72 steps (range 1–10).

### Trial-wise preprocessing and exclusions

As preregistered, we only included trials with long anticipation-to-movement screen intervals in our analyses (i.e., 150 trials per participant). For 8 (out of 58) subjects we only had usable data of 2 out of 3 runs (i.e., 100 trials), due to a lack of observations in our cells of interest for fMRI analysis (i.e., passive and active approach-avoidance choices). For consistency, we used the same data set across all analyses. Additionally, we excluded trials with poor heart rate data, or response times below 200 ms (i.e., excluding a further 304 trials (±3.7%) from the data set). For all analyses involving response times, only trials with active responses (i.e., button-presses) were used (±48.7% of the data set).

### Physiological preprocessing

Raw pulse and respiratory data were preprocessed using in-house software for removal of radio frequency artifacts and interactive visual artifact correction and peak detection (https://github.com/can-lab/brainampconverter, https://github.com/can-lab/hera). In general, MR artifacts were identified as the modulation of the signal relative to each TR and subsequently removed using deconvolution. The cleaned heart rate signals were high-pass filtered at 0.01 Hz, and low-pass filtered at 2.5 Hz. Cardiac inter beat intervals (IBI's) of the peaks were converted to beats-per-minute (BPM = 60/IBI) and baseline corrected with respect to the average heart rate (BPM) during the 1 second baseline window prior to the trial onset (i.e., $t_{baseline} = [−1, 0]$). As pre-registered, the trial-by-trial quantification of bradycardia states ($\Delta hr$) was computed by taking the average baseline-corrected heart rate across a 5–7 s time window (such that increasingly negative values reflect heart rate deceleration or bradycardia, compared to baseline). Finally, respiratory data were used for physiological noise correction of the BOLD-fMRI signal (see *Physiological noise correction* below).

### MRI data acquisition

MRI scans were acquired using a 3 T MAGNETOM PrismaFit MR scanner (Siemens AG, Healthcare Sector, Erlangen, Germany), using a 32 channel head coil. T2*-weighted BOLD-fMRI was acquired using a multiband sequence (68 axial slices, TR = 1500 ms, TE = 28 ms, flip angle = 75°, multiband acceleration factor = 4, interleaved slice acquisition, slice thickness = 2 mm, voxel size = 2 mm isotropic, phase encoding direction = A»P, bandwith = 2290 Hz/Px, echo spacing = 0.54 ms, phase partial Fourier = 7/8, FOV = 210 × 210). Additionally, a field map was acquired to correct for distortions due to structural magnetic field inhomogeneities (68 slices, TR = 435 ms, TE1 = 2.20 ms, TE2 = 4.66 ms, flip angle = 60°). Finally, one anatomical image per participant (1 mm isotropic) was acquired using a single-shot T1-weighted magnetization-prepared rapid gradient-echo sequence (MP-RAGE; acceleration factor of 2 [GRAPPA method], TR = 2300 ms, TE = 3.03 ms, flip angle = 8°, 192 sagittal slices, FOV = 256 × 256 × 192 mm).

### MRI data processing

**Preprocessing**. All raw MRI images were converted to nifti format and then preprocessed in SPM12 (Statistical Parametric Mapping; Wellcome Trust Centre for Neuroimaging, London, UK). Functional images were realigned and unwarped using the field-map based voxel displacement map, coregistered to the anatomical image using maximization-based rigid-body registration, normalized to MNI152 space (Montreal Neurological Institute), and spatially smoothed with a Gaussian kernel of 5 mm full width at half maximum. Six rigid-body realignment parameters were added to the GLMs analyses as nuisance regressors.

**Physiological noise correction**. To correct for cardiac and respiratory noise in the BOLD signal, we applied the same procedure as described by de Voogd et al.[89]. The preprocessed pulse and respiration measures were used for retrospective image-based correction (RETROICOR) of physiological noise artifacts in BOLD-fMRI data[46]. Raw pulse and respiratory data were used to specify fifth-order Fourier models of the cardiac and respiratory phase-related modulation of the BOLD signal[90], yielding 10 nuisance regressors for cardiac noise and 10 for respiratory noise. Additional regressors were calculated for heart rate frequency, heart rate variability, (raw) abdominal circumference, respiratory frequency, respiratory amplitude, and respiration volume per unit time[91], yielding a total of 26 RETROICOR regressors (https://github.com/can-lab/RETROICORplus).

### Statistics and reproducibility

**General**. All statistical analyses were performed on a sample size of 58 participants. Throughout behavioral and computational analyses, we used Bayesian mixed-effects (also 'hierarchical' or 'multi-level') models to estimate group-level (i.e., fixed) effects while controlling for

participant-level (i.e., random) differences in intercepts and slopes. All models were fitted using the {brms} and {rstan} packages in R (with Rstudio[92-96]) using 6000 samples (3000 warmup) across 4 chains, and all converged to a solution without warnings or errors (i.e., all R-hat values between 0.99 and 1.01). All models with binary dependent variables (e.g., approach/avoid choices or passive/active responses) were modeled using Bernoulli distributions (logit link), response time variables with shifted log-normal distributions (identity link), and all others with Gaussian distributions. All models followed a 'maximal' random-effects structure (i.e., by-participant random intercepts, by-participant random slopes for all within-subjects effects, and all pairwise correlations between random intercepts and slopes)[97]. Finally, for all models we used the default (i.e., weakly regularizing) {brms} priors, which are improper flat priors for group-level effects, weakly informative Student-t priors for participant-level effects (i.e., random intercepts and slopes), and LKJ-Correlation priors for random correlations.

To interpret the output of the Bayesian models, we report highest density (credible) intervals (HDIs) of the posterior parameter distributions. These are analogous but not exactly equivalent to frequentist confidence intervals. For example, a 95% HDI indicates the range that includes the 95% most probable parameter estimates (i.e., with the highest probability density), given the evidence provided by the data. All values falling within the interval have a higher probability to reflect the true (unknown) parameter estimate than those falling outside of the interval. Thus, when a large percentage of the posterior distribution is unidirectionally non-zero, this indicates a large probability of the true effect also being non-zero (in that direction). While Bayesian approaches thus allow us to make continuous statements about the probability of an effect (see e.g., Limbachia et al.[98]), to aid interpretation we report two types of statements pertaining to effects' significance. Whenever the 95% HDI *does not* include 0 (e.g., it ranges from 0.3 to 1.4), we interpret the effect to be 'significant'. Whenever the 95% HDI does, but the 90% HDI does not include 0, we interpret the effect as 'marginally significant'[99-102]. Finally, we also report effects' posterior estimates; these are simply the means of the posterior parameter distributions.

*Task effects on choices, response times, and heart rate.* We tested effects of the task (i.e., money and shocks levels, and the action context) on decision-making and heart rate ($\Delta hr$) using Bayesian mixed-effects models. For the full model specifications of the choice and response time models in {brms} syntax, see Eqs. (1) to (3) below.

Choice model:

$$
\begin{aligned}
choice \sim\ & money * shocks + ac + money : ac + shocks : ac \\
& + (1 + money * shocks + ac + money : ac + shocks : ac|pID)
\end{aligned}
\tag{1}
$$

Response time model:

$$
\begin{aligned}
response\ time \sim\ & money * shocks * choice \\
& + (1 + money * shocks * choice|pID)
\end{aligned}
\tag{2}
$$

Heart rate model:

$$
\begin{aligned}
heart\ rate \sim\ & money * shocks * ac \\
& + (1 + money * shocks * ac|pID)
\end{aligned}
\tag{3}
$$

In this notation, the term left of the tilde ('~') denotes the dependent variable and the terms to the right of the tilde denote the predictors, the terms between brackets denote the random effects structure (i.e., '1' reflects the random intercept, and the variables reflect the random slopes, all of which are modeled separately for participants ('pID') as the grouping factor), and an asterisk (*) between two variables indicates that both main effects as well as their interaction are modeled, whereas a colon (:) indicates an isolated interaction term between two variables.

**Behavioral interactions with bradycardia states.** To statistically test the effect of trial-by-trial bradycardia states on approach-avoidance choices, we ran a Bayesian mixed-effects model with the same effects specification as Eq. (1), but with trial-by-trial $\Delta hr$ as an additional continuous predictor, both as a main effect as well as interacting with all the task effects, as follows:

$$
\begin{aligned}
choice \sim\ & money * shocks + ac + money : ac + shocks : ac + \Delta hr \\
& + money : \Delta hr + shocks : \Delta hr + ac : \Delta hr + money \\
& : shocks : \Delta hr + money : ac : \Delta hr + shocks : ac : \Delta hr \\
& + (1 + money * shocks + ac + money : ac + shocks : ac + \Delta hr \\
& + money : \Delta hr + shocks : \Delta hr + ac : \Delta hr + money : shocks : \Delta hr \\
& + money : ac : \Delta hr + shocks : ac : \Delta hr|pID)
\end{aligned}
\tag{4}
$$

We similarly tested the effect of bradycardia states on response times, adding $\Delta hr$ as main and interaction effects to the response time model in Eq. (2):

$$
\begin{aligned}
response\ time \sim\ & money * shocks * choice * \Delta hr \\
& + (1 + money * shocks * choice * \Delta hr|pID)
\end{aligned}
\tag{5}
$$

**Computational modeling of bradycardia-state interactions**
Using the same mixed-effects (i.e., hierarchical) Bayesian modeling approach, we performed computational modeling to test our hypotheses regarding the three mechanisms through which freezing may affect computations underlying approach-avoidance choices.

**Model specification.** First, we specified a base model estimating the value of the decision (decision value; DV, ranging $[-\infty, +\infty]$) on each trial, accounting for task-induced effects and a main effect of bradycardia on choice. The DV function of this base model consists of 8 free parameters, estimating the average approach rate ($\beta_0$), the effects of money ($\beta_m$), shocks ($\beta_s$), action context ($\beta_{ac}$), and bradycardia ($\beta_{hr}$) on choice, and the two-way interaction effects between money and shocks ($\beta_{m:s}$), money and action context ($\beta_{m:ac}$), and shocks and action context ($\beta_{s:ac}$):

$$
\begin{aligned}
DV_{Base} = & \beta_0 + \beta_m \cdot m + \beta_s \cdot s + \beta_{m:s} \cdot m \cdot s + \beta_{ac} \cdot ac + \beta_{hr} \\
& \cdot \Delta hr + \beta_{m:ac} \cdot m \cdot ac + \beta_{s:ac} \cdot s \cdot ac
\end{aligned}
\tag{6}
$$

Where $m$ and $s$ indicate the potential money amounts and shocks, $\Delta hr$ reflects the change in heart rate relative to the pre-trial baseline, and $ac$ is a sum-to-zero coded dummy variable indicating the action context:

$$
ac = \begin{cases} -1, & action\ context = active \\ 1, & action\ context = passive \end{cases}
\tag{7}
$$

Specifically, $\beta_{ac}$ captures the variance in approach-avoidance choices due to the participants' tendency to respond passively ($ac > 0$) vs. actively ($ac < 0$).

Note that this model has the same parameterization as Eq. (1), except for the additional inclusion of a main effect of heart rate on choice. This additional term was included to improve the interpretability of the interaction between $\Delta hr$ and other terms introduced in the freezing models (see below). To fit this model to the choice data, the trial-by-trial DVs were transformed to choice probabilities (ranging $[0,1]$) through the logistic function:

$$
p(approach) = \frac{1}{1 + e^{-(DV)}}
\tag{8}
$$

Next, building from this base-model parameterization, we formalized three new freezing models that each add a single term to the base DV

**Table 1 | Group-level parameter estimates used to compute trial-wise decision values (DV) for each model**

| Model | $\beta_0$ | $\beta_m$ | $\beta_s$ | $\beta_{m:s}$ | $\beta_{ac}$ | $\beta_{hr}$ | $\beta_{m:ac}$ | $\beta_{s:ac}$ | $\beta_{s:hr}$ | $\beta_{ms:hr}$ | $\beta_{ac:hr}$ |
|---|---|---|---|---|---|---|---|---|---|---|---|
| Base | 0.73 | 1.54 | −1.10 | 0.41 | 0.07 | 0.05 | −0.28 | 0.09 | – | – | – |
| AV | 0.73 | 1.54 | −1.11 | 0.40 | 0.07 | 0.04 | −0.28 | 0.09 | 0.07 | – | – |
| VC | 0.73 | 1.55 | −1.11 | 0.40 | 0.07 | 0.03 | −0.28 | 0.09 | – | −0.09 | – |
| AI | 0.73 | 1.54 | −1.10 | 0.40 | 0.07 | 0.05 | −0.28 | 0.09 | – | – | 0.05 |

Note: As described above, the base model did not include any bradycardia interaction term, and the freezing models only included one bradycardia interaction term each (three right-most columns).
*AV* aversive value, *VC* value comparison, *AI* action invigoration, *m* money, *s* shocks, *ac* action context, *hr* heart rate (Δhr), *ms* differential money-shock level, *colons* (:) depict interaction terms between two variables.

function, corresponding to the three hypothesized mechanisms of bradycardia-state interactions underlying approach-avoidance:

Aversive value (AV):

$$DV_{Base} + \beta_{s:hr} \cdot s \cdot \Delta hr \qquad (9)$$

Value comparison (VC):

$$DV_{Base} + \beta_{ms:hr} \cdot \Delta ms \cdot \Delta hr \qquad (10)$$

where the $\Delta ms$ variable is computed as the difference between the money and shock levels (i.e., $\Delta ms = money - shocks$).

Action invigoration (AI):

$$DV_{Base} + \beta_{ac:hr} \cdot ac \cdot \Delta hr \qquad (11)$$

The trial-by-trial DVs of the base model and these three additional models were subsequently used in the model-based fMRI analysis.

**Model fitting procedure.** While we initially planned to fit these computational models using MLE (maximum likelihood estimation) and on the subject level (rather than hierarchically), we instead opted for a hierarchical (Bayesian) approach because this tends to estimate group and participant-level parameters more reliably, compared to more traditional 'flat' (i.e., non-hierarchical) model-fitting procedures[103]. For fitting these models we used the same settings for other BMMs as specified in the *Statistics and Reproducibility – General* section (i.e., maximal random-effect structure, default uninformed priors, all continuous predictors mean-centered and scaled across subjects).

**Computation of trial-by-trial decision values (DV).** To compute the trial-by-trial DVs for each of the models (per participant), we multiplied the trial-wise data (i.e., money and shock levels, action context, and heart rate) with the group-level parameter estimates obtained from the fitted Bayesian models described in the *Model specification* section (see for example also Zorowitz et al.[68]). We used the group-level parameter estimates for this rather than participant-level estimates, because – following previous reports[48,104,105]—we wanted to investigate the neural correlate of the group-level behavioral effects that the models estimate, not in individual differences that may obscure the common underlying cognitive process. Table 1 displays an overview of the exact parameter values used to generate the trial-wise DVs per participant for each of the models (obtained from the model fitting procedure). The associated behavioral predictions are drawn in main text Fig. 5B.

**fMRI analysis**
Since our analyses of interest only included trials with long anticipation-to-movement screen intervals (AMI), the remaining trials with shorter AMI's (± 20% of trials per participant) were modeled in separate regressors of no interest so as to exclude them from the implicit baseline. Unless specified otherwise, all regressors were temporally convolved with the hemodynamic response function (HRF) included in SPM12 (Statistical Parametric Mapping; Wellcome Trust Centre for Neuroimaging, London, UK).

Additionally, we included six movement parameter regressors (3 translations, 3 rotations), 26 RETROICOR regressors, high-pass filtering (1/ 128 Hz cutoff), and AR(1) serial correlation corrections into all models. First-level contrast maps were entered into second-level one-sample t-tests.

**Correction for multiple comparisons.** For analyses within regions of interest (ROI) we use peak-level correction ($p < .05$ FWE-corrected), as preregistered. While for whole-brain level analyses we preregistered the same, we think this would not do justice to the clustered patterns of activation that we observe for our non-ROI contrasts (e.g., approach vs. avoid, passive vs. active, and parametric analyses of money and shock levels). Since the PAT is a new task in the fMRI context, we want to be transparent and report on any significant clusters that appear in any of these contrasts. This will facilitate comparison of our results to data of previous studies using comparable experimental paradigms[32,33,36,42,52,106]. Therefore, for whole-brain statistics only, we will report activations that are significant at the cluster level ($p < .05$ FWE-corrected, at an initial cluster-forming threshold of $p < .001$ uncorrected). Note that this will not in any way affect the conclusions of our main hypotheses concerning the bradycardia-state interactions underlying approach-avoidance, since those are tested through ROI analyses (using peak-level correction as planned).

**ROI definition.** We preregistered various regions of interest (ROIs) for our fMRI analysis. For reward magnitude effects on BOLD we defined ROIs that have traditionally been associated with reward processing, namely the vmPFC and ventral striatum[107,108]. For threat magnitude effects we defined regions that have traditionally been associated with fear, arousal, and the salience network; namely the amygdala, PAG, ACC, and BNST[35,39,109–112]. For approach-avoidance decisions, we defined the ventral striatum, amygdala, and anterior cingulate as ROIs for approach (vs. avoidance), and the frontal gyrus and anterior insula for avoidance (vs. approach). These regions were based on their previously implied role in approach-avoidance, risky decision-making, reward and threat anticipation, and risk avoidance[30,107,113,114].

For the amygdala, vmPFC, and anterior cingulate regions we used the Automated Anatomical Labeling 3 (AAL3)[115] atlas (vmPFC consisting of the frontal medial orbital, rectus, and OFC medial subregions, and ACC consisting of the sup-ACC, pre-ACC, sub-ACC, and Cingulate_Mid subregions). For the ventral striatum we used an independent mask created previously[116], because the ventral striatum consists of a combination of different anatomical regions (e.g., nucleus accumbens, part of the caudate nucleus). For the BNST we used a mask provided by Avery et al.[117]. Finally, for the PAG we used a mask based on manual segmentations from Lojowska et al.[118]. thresholded at a value of .25 (following Neubert et al.[119]).

**Task-based analysis.** To investigate the neural correlates of passive and active approach-avoidance choices, we created a first-level model containing 4 box car regressors during the anticipation screen (passive-approach, active-approach, passive-avoid, active-avoid; 0.5–7 s duration), two stick function regressors for the movement screen (passive, active response), and three box car regressors for the outcome screen (money, shocks, neutral; 1.5 s duration).

To investigate the neural correlates of reward and threat anticipation, we additionally created a model with a single box car regressor for each anticipation screen, modulated by three parametric regressors: money level, shock level, and their interaction (i.e., multiplying the demeaned money and shock regressors with each other). All parametric regressors were orthogonalized with respect to the unmodulated trial regressor per subject (i.e., demeaned)[50]. This model also contained regressors for the movement and outcome screens as specified for the previous model.

**Finite impulse response (FIR) analysis.** To investigate the time course of the BOLD response during the anticipation screen, we fit a finite impulse response model. In this model, activation during (and shortly after) the anticipation screen was estimated using a model that included 8 time bins (TR = 1.5 s) starting one time bin before the onset of the stimuli ($-1.5$ s) and ending two time bins after the anticipation screen (10.5 s). This way, bin numbers 2 to 5 (0–6 s) always fell within the anticipation screen. This first-level model makes no assumptions regarding the shape of the HRF and yields independent response estimates for each time bin. All additional preprocessing (regarding nuisance regressors, smoothing, etc.) was performed like discussed previously (see above). To analyze the time courses, we extracted the subject-wise β weights from three anatomical ROIs; the bilateral amygdala, ventral striatum, and vmPFC (see *ROI definition* section above). All responses were baseline corrected relative to the response in bin 1 and analyzed using 3-way repeated measures ANOVAs (per ROI) with within-subjects factors time (bin 2 to 5), choice (approach/avoid), and response (active/passive). Since for all models the assumption of sphericity was violated, we applied Greenhouse-Geisser correction to all degrees of freedom (DFs) and p-values, and we report partial $\eta$-squared ($\eta_p^2$) effect-size estimates where appropriate. Significant effects were followed up using two-tailed paired t-tests.

**Model-based analysis.** To test the neural correlates of our three hypothesized mechanisms of bradycardia-state interactions with computations underlying approach-avoidance arbitration, we used model-based fMRI analysis. Specifically, we extracted—per participant—the trial-by-trial DVs of the base computational model (Eq. 6) and the DVs of the three freezing models (Eqs. 9, 10, and 11). We then used these trial-by-trial values as parametric regressors in a single first-level GLM per participant. More specifically, each first-level model had a single box-car regressor for the anticipation screen, and four corresponding parametric regressors (one for the base model, and one for each of the three freezing models). The first parametric regressor contained the raw (standardized) trial-by-trial DVs of the base model. However, to account for high correlations between the base regressor and the regressors of each of the three freezing models (i.e., Spearman correlations for all freezing models with the base model across subjects were approximately $M_{Rs} = \pm 0.998$, $SD_{Rs} = \pm 0.0006$), we orthogonalized the regressors of each of the freezing models with respect to the base regressor by computing difference scores: For each of the freezing models we computed the difference in trial-wise DV predictions compared to the base model DVs, such that (e.g., for the AV model) $DVdiff_{AV} = DV_{Base} - DV_{AC}$. This approach has previously been used to deal with correlated model-based parametric regressors[47–49]. Indeed, after computing difference scores the correlations between regressors were greatly reduced ($M_{Rs\_AV} = -0.035$, $SD_{Rs\_AV} = 0.165$; $M_{Rs\_VC} = -0.088$, $SD_{Rs\_VC} = 0.249$; $M_{Rs\_AI} = 0.003$, $SD_{Rs\_AI} = 0.081$). As such, these difference regressors reflect a difference in trial-by-trial predictions of each freezing model with respect to the base model: positive difference scores reflect higher value assigned by the base model than the freezing model (i.e., the freezing model predicts relatively more avoidance), whereas negative difference scores reflect lower value assigned by the base model compared to the freezing model (i.e., the freezing model predicts relatively more approach). Importantly, because each freezing model only deviates from the base model by a single term (e.g., $\beta_{s:hr}$ for the AV model), any difference in the predicted decision value, as well as its

correlate with BOLD-fMRI, is driven by this additional term. Finally, because the difference regressors of all three freezing models were entered together into the same first-level model, the BOLD correlations for each regressor reflect the unique contribution of that regressor, while controlling for the contribution of other regressors. Any shared explained variance between freezing-model regressors will thus be assigned to the error term[50].

## Reporting summary
Further information on research design is available in the Nature Portfolio Reporting Summary linked to this article.

## Data availability
The source data supporting the findings of this study are available in the Radboud Data Repository (data.ru.nl) under the identifier di.dccn.DSC_3023009.03_522, at https://doi.org/10.34973/tvyt-h588.

## Code availability
The code used to generate the findings of this study are available in the Radboud Data Repository (data.ru.nl) under the identifier di.dccn.DSC_3023009.03_522, at https://doi.org/10.34973/tvyt-h588.

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

## Acknowledgements

This work was supported by a Consolidator Grant from the European Research Council (DARE2APPROACH; ERC_CoG – 2017_772337) awarded to K.R., and also supporting F.H.K., L.D.V., and A.H.

## Author contributions

Conceptualization, F.H.K., L.D.V., A.M.H., J.X.O., F.K., B.F., and K.R.; Methodology. F.H.K., L.D.V., J.X.O., B.F., and K.R.; Software, F.H.K. and L.D.V.; Formal Analysis, F.H.K.; Investigation, F.H.K.; Writing – Original Draft, F.H.K., L.D.V., and K.R.; Writing – Review & Editing; F.H.K., L.D.V., A.M.H., J.X.O., F.K., B.F., and K.R.; Supervision, L.D.V., B.F., and K.R.; Funding Acquisition, K.R.

## Competing interests

The authors declare no competing interests.
