## [Peer review file · Communications Biology]

Reviewers' comments:

Reviewer #1 (Remarks to the Author):

The manuscript offers an interesting and quite unique study looking at the relationship between freezing behaviour (indexed by bradycardia – deceleration of heart rate) and approach-avoidance decision-making. The authors find bradycardia is linked to increased shock-induced avoidance, i.e., trials with stronger bradycardia and shock level had increased avoidance, suggesting that bradycardia relates to aversive value. This effect was negatively correlated with BOLD activation in the amygdala which the authors attribute to an increase in attentional processing. They also find that bradycardia is related to the comparison of valuable and threatening options; trial-by-trial bradycardia was related to avoidance when reward was equal or less than the threat on the trial, whereas bradycardia related to approach when reward outweighed the threat on the trial. These changes were linked to BOLD signals in dACC/SMA, which the authors suggest reflect the integration of the subjective value of outcomes. Ultimately the study shows the relevance of psychophysiological states such as heart rate deceleration in decision processes involved in making approach-avoidance choices under threat.

Methodologically, the study is robust and thorough – the use of Bayesian hierarchical modelling is a real strength as it maximises statistical power, accounting for both within-subject and between-subject effects. Further, it is commendable that the authors preregistered their analyses, although some of the preregistration is a little vague and leaves room for post-hoc changes (e.g., stating “regions such as...”). I have a few comments that may be worth addressing in a revision, although these are primarily clarifications rather than substantial changes.

- Whilst the work is unique, it could be more clearly de-lineated from the authors previous and highly related behavioural work (Klaassen et al, 2021, Scientific Reports). I.e., it is not clearly outlined what has already been shown by the research on the topic, and thus what the current manuscript is seeking to add specifically in terms of behaviour (e.g., is it to look just for neural correlates?). Furthermore, it should be clearer whether the current study replicates key findings from previous work.
- In Figure 1A you present the neurocomputational effects of freezing, but this isn't intuitive, and the predicted hypotheses could be more clearly articulated. Further, what is the significance of passive vs active approach/avoid here? How does this affect the proposed mechanism?
- In Figure 2B would be useful to see error bars (shaded perhaps) in the heart rate deceleration figure.
- In Figure 4 authors should make their mention of 'marginally significant' more transparent by reporting exact values rather than '#' and make it more explicit what constitutes 'significant' here (this is mentioned in the text, but it would be useful for it to be mentioned in the figure caption too). This should also perhaps rephrased, given that “significance” is typically associated with frequentist approaches where these results are based on Bayesian posterior HDIs.
- Figure 5 requires clear definition of error bars.
- The justification for the regions chosen as ROIs should be clearer and backed up with literature. Many of these make intuitive sense (e.g., striatum = reward), but there could be more explicit justification for these with reference to prior literature specifically in the approach/avoidance domain.
- Were p-values corrected for multiple comparisons across ROIs? I.e., did the analyses use separate

ROIs with SVC for each region, with resulting p-values corrected using e.g., FDR? Or was a single ROI used for all regions, allowing for SVC across all constituent regions?

- The authors offer interpretations of the key findings separately, but this could be further developed to integrate both the aversive value and value comparison findings to offer more mechanistic view of the role of bradycardia/freezing in these neurocomputations. What are the implications if, as you speculate, the bradycardia/freezing is an epiphenomenon?
- The authors do not report/explore relationships with questionnaire measures which seems a shame as the authors are specifically interested in links with psychopathology. The study may well be underpowered for looking at links with individual differences in measures external to the task, but these may allow for some interesting exploratory analyses.
- Relatedly, the authors report collecting pupil diameter measures, but this is not mentioned again. Were these analysed? I wouldn't necessarily expect the authors to run this additional analysis if they have reason not to, but it would be at least worth explaining in the text.
- In general, the manuscript references previous literature appropriately, however the literature on interoception is highly relevant and not mentioned. Secondly the authors mention a need for high temporal precision MEG work, but do not mention related work in approach-avoidance paradigms (e.g., Khemka et al., 2017).

Reviewer #2 (Remarks to the Author):

Dear Editor,

I read the article written by Klaassen and colleagues titled "The neurocomputational link between defensive cardiac states and approach-avoidance arbitration under threat".

I found, the study well conducted, and very interesting. Therefore, this manuscript can be of interest for the readers of the Communication Biology. I have to say that I do not feel sufficient expert for the modelling applied in the study and therefore I leave this judgement to other reviewers or the editor.

The study was in general clearly described and I do not have many comments, but only a few.

General Comments

1. While reading the abstract, I missed reading the goal as well as some details about the method.
2. The (post-hoc) stars in Fig. 2 are unclear to me, at least it is unclear to which comparisons they refer to.
3. How did the "researcher made sure participants fully understood the task"?
4. The properties of the aversive US are not precisely described. The shock lasted 200 ms, was this stimulation continuous or was it a train of pulses? To me, it is unclear what is meant by "the intensity varied in 10 steps". What was precisely the standardized calibration procedure? Were the intensities of the five shocks pre-defined?
5. The two USs differed in regard to several aspects, e.g. primary vs. secondary reinforcer, the duration. Did the authors collect any subjective ratings for these two types of stimuli? Were they comparable in regards to their arousal?
6. In the discussion, there is no mention about the limitations of the study. The only one is the slow

imaging method.

Minor comments

1. I am not sure the term “fearful avoidance” is the most appropriate. Avoidance can be elicited by a threat or by the feeling of fear, but it cannot be fearful per sé.
2. Specify the size of the electrodes used.
3. Did the authors apply any filter for the physiological responses?
4. Fig.3 and Fig 5. depict correlations. Close to the scale, I found the abbreviation “uncorr.” misleading. It refers to uncorrected, but because of the correlations depicted in the picture I first thought that it means uncorrelated.

Reviewer #1 (Remarks to the Author):

The manuscript offers an interesting and quite unique study looking at the relationship between freezing behaviour (indexed by bradycardia – deceleration of heart rate) and approach-avoidance decision-making. The authors find bradycardia is linked to increased shock-induced avoidance, i.e., trials with stronger bradycardia and shock level had increased avoidance, suggesting that bradycardia relates to aversive value. This effect was negatively correlated with BOLD activation in the amygdala which the authors attribute to an increase in attentional processing. They also find that bradycardia is related to the comparison of valuable and threatening options; trial-by-trial bradycardia was related to avoidance when reward was equal or less than the threat on the trial, whereas bradycardia related to approach when reward outweighed the threat on the trial. These changes were linked to BOLD signals in dACC/SMA, which the authors suggest reflect the integration of the subjective value of outcomes. Ultimately the study shows the relevance of psychophysiological states such as heart rate deceleration in decision processes involved in making approach-avoidance choices under threat.

Methodologically, the study is robust and thorough – the use of Bayesian hierarchical modelling is a real strength as it maximises statistical power, accounting for both within-subject and between-subject effects. Further, it is commendable that the authors preregistered their analyses, although some of the preregistration is a little vague and leaves room for post-hoc changes (e.g., stating “regions such as...”). I have a few comments that may be worth addressing in a revision, although these are primarily clarifications rather than substantial changes.

We thank you for your compliments and for the elaborate feedback.

1. Whilst the work is unique, it could be more clearly de-lineated from the authors previous and highly related behavioural work (Klaassen et al, 2021, Scientific Reports). I.e., it is not clearly outlined what has already been shown by the research on the topic, and thus what the current manuscript is seeking to add specifically in terms of behaviour (e.g., is it to look just for neural correlates?). Furthermore, it should be clearer whether the current study replicates key findings from previous work.

We agree that we can outline this better. To clarify, the Klaassen et al. (2021) study was a first behavioral exploration of the relation between defensive freezing states and approach-avoidance decision-making. In that study, we first observed a potential relation between freezing states (indicated by bradycardia and bodily immobility) and the value-based processes underlying passive vs. active approach-avoidance decision-making on the subject level. However, the question remained what underlying mechanisms give rise to this relationship.

The current study is a follow-up study to investigate the neural correlates of the potential relationship between freezing states and approach-avoidance decision-making. Additionally, we here more concretely focused on three pre-defined hypothesized routes. These three routes (outlined in Livermore et al., 2021) describe three potential neurocognitive mechanisms through which freezing states may affect approach-avoidance decision-making: 1) aversive value, 2) value comparison, and 3) the switch to action, now renamed as ‘action invigoration’. The current study aimed to directly test these three neurocognitive routes on the trial-by-trial level in both brain and behavior.

The current study partially replicates Klaassen et al. (2021). Generally, we again find a relationship between value-based processing and freezing states (indexed by bradycardia). However, this time we more specifically show that bradycardia states are associated with the effect of aversive value and value comparison, and that this already happens on the trial-by-trial level. That is, we now find that approach-avoidance decision-making is moderated by the *current* psychophysiological freezing-like state, and that the amygdala and dACC are involved in these interactions. Additionally, we replicate that these states are associated with faster response times (shown before by Klaassen et al. 2021 and others).

We clarified this in the **Introduction** and **Discussion**:

In the **Introduction** (line 72 - 83):

“Interestingly, in a recent behavioral study (Klaassen et al., 2021) we found evidence in line with a role of freezing states in instrumental decision-making on the subject level. In this study, bradycardia was associated with value integration of reward and threat during approach-avoidance arbitration, depending on the action context. However, this relationship occurred on the subject level, and so it remains unknown how transient defensive cardiac states might affect value-based computations and underlying neural circuits on a momentary (trial-by-trial) basis. This knowledge is critical to provide starting points for optimizing interventions aiming at improved decision-making under threat in health and anxiety.

To address this knowledge gap, we a priori formulated three potential mechanisms by which freezing states could impact approach-avoidance arbitration under threat (corresponding to routes 1 – 3 in Figure 1A), previously published in Livermore et al. (2021)”

In the **Discussion** (line 497 - 499):

“Whereas previously such association between computations of subjective outcome value and bradycardia states was found across participants (Klaassen et al., 2021), we now show this on a trial-by-trial level. We thus extend previous work (Klaassen et al., 2021), illustrating how the current physiological state of the decision maker may inform approach-avoidance arbitration.”

And line 529:

“However, we did find a trial-by-trial relationship between bradycardia and faster response times, replicating previous work (Gladwin et al., 2016; Klaassen et al., 2021; Löw et al., 2015; Roelofs, 2017; Wendt et al., 2017)”

- 2. In Figure 1A you present the neurocomputational effects of freezing, but this isn't intuitive, and the predicted hypotheses could be more clearly articulated. Further, what is the significance of passive vs active approach/avoid here? How does this affect the proposed mechanism?**

To make the figure more intuitive we now improved the illustration. Figure 1A presents the three hypothesized neurocognitive routes proposed in Livermore et al. (2021) and based on Klaassen et al. (2021). To clarify these hypotheses, we now numbered the routes and refer to these numbers when discussing the three routes in the figure caption and general text.

The significance of passive vs active approach/avoid is the following. The extent to which freezing states are related to these computations follows from changes in subsequent approach-avoidance choices, and the extent to which these are active or passive. Specifically, changes in route 1 and 2 affect the proportion of approach (vs. avoidance) choices as a function of threat and reward-threat differences (respectively), whereas changes in route 3 would affect the proportion of active vs. passive approach-avoidance choices. Thus, the significance of distinguishing between passive vs. active approach-avoidance choices follows from the hypothesized computation in route 3.

We hope this information now follows more intuitively from the improved Figure 1A (see below).

A Neurocomputational effects of freezing states on approach-avoidance arbitration

B Passive-active Approach-avoidance Task (PAT)

“Figure 1. Theoretical and experimental paradigm – outline of hypothesized mechanisms of freezing-state effects on approach-avoidance, and trial design of the Passive-active Approach-avoidance Task (PAT). A) Visualization of the theoretical framework in which we hypothesize three routes through which freezing states may affect the computations underlying approach-avoidance arbitration under threat; processing of aversive value in the amygdala and periaqueductal gray (AMY, PAG; route 1), value comparison in the dorsal anterior cingulate cortex (dACC; route 2), and action invigoration in the perigenual ACC (pgACC; route 3). This model hypothesizes that freezing effects on the computations in routes 1 and 2 affect the proportion of approach (vs. avoidance) choices as a function of threat (route 1) and reward-threat comparison (route 2), and the proportion of active vs. passive approach-avoidance choices (route 3). See main text for details and rationale. Figure was adapted with permission from Livermore et al., 2021. vmPFC: ventromedial prefrontal cortex; vStr: ventral striatum. B) To test this model, we developed an experimental task in which participants (white square) have to approach or avoid targets (gray circle) that are associated with varying reward and threat magnitudes (ranging 1-5 euro/shocks and indicated by green coins/lightning bolts respectively). After an initial anticipation screen (duration of 6 – 7 s for 80% of the trials) participants could indicate their choice during the target movement window (700 ms) by positioning themselves (i.e., the white square) on the same location as where the target was moving (approach) or on the other location (avoid). If participants approached, there was a large probability to receive either the indicated number of shocks (40%) or amount of money (40%), and a small probability to receive nothing (20%). If participants avoided, there was a large probability to receive nothing (80%), and smaller probabilities to receive shocks (10%) or money (10%). Moreover, the target movement direction during the movement window was manipulated in two action contexts (i.e., movement towards or away from the player, 50% of all trials each), such that participants could always either actively approach/passively avoid, or passively approach/actively avoid. Participants were fully instructed about these task conditions (including the outcome probabilities). Dashed arrows were not present in the actual task.”

We have now also more explicitly linked the results in Figure 4 to each of the three routes by numbering the corresponding panels:

And by clarifying the corresponding figure caption:

“Figure 4. Comparison of observed and model-predicted bradycardia-state interactions underlying approach-avoidance choices. Observed (top) and model-predicted (bottom) effects on choice for the three hypothesized routes (1: aversive value, 2: value comparison; 3: action invigoration). Specifically, we plot the interactions between bradycardia and A) the number of shocks, B) the money-shock difference Δhr , and C) the action context. Model-based plots also display the posterior distributions of the interaction coefficients, with the 90% HDI shaded in red, and the 95% HDI in orange. For plotting purposes only, we created conditions with stronger bradycardia (low Δhr) and weaker bradycardia (high Δhr ; respectively containing the trials with the 33% lowest vs highest Δhr values). Model predictions are plotted as predicted approach probabilities. Error bars indicate ± 1 SEM; asterisks (*) indicate ‘significant’ effects (i.e., HDI_{95%} excludes 0), hash icons (#) indicate ‘marginally significant’ effects (i.e., HDI_{90%} excludes 0), n.s.: not ‘significant’ (i.e., HDI_{90%} includes 0); m: money; s: shocks.”

3. In Figure 2B would be useful to see error bars (shaded perhaps) in the heart rate deceleration figure.

We agree and have added shaded error bars to the heart rate plot in Figure 2B:

4. In Figure 4 authors should make their mention of ‘marginally significant’ more transparent by reporting exact values rather than ‘#’ and make it more explicit what constitutes ‘significant’ here (this is mentioned in the text, but it would be useful for it to be mentioned in the figure caption too). This should also perhaps be rephrased, given that “significance” is typically associated with frequentist approaches where these results are based on Bayesian posterior HDIs.

We have added the definition of (marginally) ‘significant’ effects in the captions of all figures containing significance indicators. We think this improves transparency. However, to avoid cluttering the figures, and given that the accompanying text contains the precise statistical information for all effects, we opted to keep the */# indicators in the figures. We hope you agree with this compromise.

We acknowledge that using frequentist terminology like ‘significant’ might be confusing. Originally, we opted for this terminology because even though the Bayesian estimation method differs from the Frequentist approach, the type of statistical inference we make in our study is highly similar (i.e., an effect is ‘significant’ if a certain statistical threshold – here with respect to the credible interval – is

reached). Thus, we think that the intuitive meaning of ‘significance’ as used in our manuscript is accurate. One potential alternative for the Frequentist ‘significance’ terminology that conveys the same meaning is to use terms of ‘credibility’ or ‘probability’ (see e.g., Hespanhol et al., 2019; Quandt et al., 2021). However, we fear that these terms may in turn be unintuitive or confusing for most readers who are not (yet) familiar with this approach. Considering these points and given that we have now included more explicit definitions in the text/figure captions for transparency (see below), we hope that you agree to keep the ‘significance’ terminology. We are happy to reconsider if you disagree with our argumentation.

For Figure 2:

“Figure 2. Task effects on choice behavior and heart rate. A) Higher money and shock levels led to more approach vs avoid choices, respectively. **B)** We observed a significant average heart rate deceleration during the anticipation screen relative to the 1 s pre-trial baseline indicative of a freezing-like bradycardia state, which was numerically but not significantly more pronounced in avoid (compared to approach) trials (see main text for statistics). Trial-by-trial bradycardia (Δhr) was quantified as the average baseline-corrected heart rate across a 5-7 s time window relative to the anticipation screen onset, such that lower (relative to higher) Δhr values indicate stronger bradycardia. **C)** Stronger trial-by-trial bradycardia was associated with faster response times (for illustration purposes, individual dots reflect RT and Δhr values aggregated separately for money and shock levels, and approach and avoid choices). Moreover, we observed faster response times for avoid compared to approach decisions. Error bars indicate ± 1 SEM. Gray-white striped shaded area in **B** reflects partial overlap between anticipation and target movement screens across different trials (i.e., movement window onset was uniformly jittered between 6 – 7 s relative to the anticipation screen onset). BPM: beats per minute; Ap.: approach; Av.: avoid; asterisks (*) indicate ‘significant’ effects (i.e., HDI_{95%} excludes 0) of money and shocks on choice **(A)** and heart rate and choice on response times **(B)**.”

For Figure 4:

“Figure 4. Comparison of observed and model-predicted bradycardia-state interactions underlying approach-avoidance choices. Observed (top) and model-predicted (bottom) effects on choice for the three hypothesized routes (1: aversive value, 2: value comparison; 3: action invigoration). Specifically, we plot the interactions between bradycardia and **A)** the number of shocks, **B)** the money-shock difference Δms , and **C)** the action context. Model-based plots also display the posterior distributions of the interaction coefficients, with the 90% HDI shaded in red, and the 95% HDI in orange. For plotting purposes only, we created conditions with stronger bradycardia (low Δhr) and weaker bradycardia (high Δhr ; respectively containing the trials with the 33% lowest vs highest Δhr values). Model predictions are plotted as predicted approach probabilities. Error bars indicate ± 1 SEM; asterisks (*) indicate ‘significant’ effects (i.e., HDI_{95%} excludes 0), hash icons (#) indicate ‘marginally significant’ effects (i.e., HDI_{90%} excludes 0), n.s.: not ‘significant’ (i.e., HDI_{90%} includes 0); m: money; s: shocks.”

5. Figure 5 requires clear definition of error bars.

We have added the error bar definition to the Figure 5 caption:

“Figure 5. Neural circuits underlying the link between bradycardia states and value-based computations. A) For the base model, we find positive correlations between model-extracted decision values (DVs, indicating the value of approaching vs. avoiding) and BOLD activity in the ventral striatum (vStr) and ventromedial prefrontal cortex (vmPFC), delineating a similar neural circuit as was observed

from the task-based analysis (see Fig. 3). **B)** DVs estimated from the aversive value (AV) model were, relative to the base model, negatively correlated with BOLD activity in the amygdala (top). A follow-up analysis revealed that this negative relationship was specific to conditions in which AV predicted more avoidance, compared to the base model (i.e., $DV_{diff_{AV}} > 0$; bottom). **C)** DVs estimated from the value comparison (VC) model were, relative to the base model, positively correlated to BOLD in a large whole brain-significant cluster in the dorsomedial prefrontal cortex (dmPFC), spanning the SMA and dACC (top). Follow-up analysis revealed that this relationship was specific to conditions in which VC predicted more approach, compared to the base model (i.e., $DV_{diff_{VC}} < 0$; bottom). $DV_{diff_{AV/VC}}$: trial-wise difference scores of the DVs of AV/VC models relative to the base model. Positive/negative $DV_{diff_{AV/VC}}$ values indicate stronger predicted probability to avoid/approach by AV/VC models compared to the base model. **Error bars indicate ± 1 SEM.** All labelled areas are significant at $p < .05$ FWE-corrected.”

6. The justification for the regions chosen as ROIs should be clearer and backed up with literature. Many of these make intuitive sense (e.g., striatum = reward), but there could be more explicit justification for these with reference to prior literature specifically in the approach/avoidance domain.

We thank you for pointing this out and we have added additional justification for the ROI definitions (see **Methods**, line 901 - 919):

“We preregistered various regions of interest (ROIs) for our fMRI analysis. For reward magnitude effects on BOLD we defined ROIs that have traditionally been associated with reward processing, namely the vmPFC and ventral striatum (e.g., Hiser & Koenigs, 2018; Rolls et al., 2008). For threat magnitude effects we defined ROIs that have traditionally been associated with threat, arousal, and the salience network; namely the amygdala, PAG, ACC, and BNST (e.g., Hulsman et al., 2021; Klumpers et al., 2017; Kolling et al., 2014; Mobbs et al., 2007, 2009; Roy et al., 2014). For approach-avoidance decisions, we defined the ventral striatum, amygdala, and anterior cingulate as ROIs for approach (vs. avoidance), and the frontal gyrus and anterior insula for avoidance (vs. approach). These regions were based on their previously implied role in approach-avoidance, risky decision-making, reward and threat anticipation, and risk avoidance (e.g., Christopoulos et al., 2009; Hiser & Koenigs, 2018; Kirlic et al., 2017; van Duijvenvoorde et al., 2015).

For the amygdala, vmPFC, and anterior cingulate regions we used the Automated Anatomical Labeling 3 (AAL3)(Rolls et al., 2020) atlas (vmPFC consisting of the frontal medial orbital, rectus, and OFC medial subregions, and ACC consisting of the sup-ACC, pre-ACC, sub-ACC, and Cingulate_Mid subregions). For the ventral striatum we used an independent mask created previously (Piray et al., 2017), because **it the ventral striatum** consists of a combination of different anatomical regions (e.g., nucleus accumbens, part of the caudate nucleus). For the BNST we used a mask provided by Avery and colleagues (Avery et al., 2014). Finally, for the PAG we used a mask based on manual segmentations from Lojowska and colleagues (Lojowska et al., 2015) thresholded at a value of .25 (following Neubert and colleagues; Neubert et al., 2015).”

7. Were p-values corrected for multiple comparisons across ROIs? I.e., did the analyses use separate ROIs with SVC for each region, with resulting p-values corrected using e.g., FDR? Or was a single ROI used for all regions, allowing for SVC across all constituent regions?

First, we would like to clarify our approach. All ROI tests used separate bilateral masks per region. For example, for testing the approach-avoidance effect in Figure 3B we performed an SVC (FWE-corrected) test for a mask containing the bilateral amygdala, and a separate test for a mask for the bilateral ventral striatum.

We would like to point out that all our ROI analyses were hypothesis-driven. First, this follows from our pre-registration. Secondly, for our main research question, the key regions – and corresponding tests – follow directly from the model previously published by Livermore et al. (2021). Because these tests were driven by theory, our analyses only contain a small number of ROIs per hypothesis. Namely, Routes 2 and 3 both only contain one ROI (i.e., the dACC and pgACC respectively). However, Route 1 indeed contains more than one ROI (i.e., the amygdala and periaqueductal gray) and correction for multiple comparison could be applied here.

To mitigate your concern regarding multiple comparisons, we performed a simple but conservative Bonferroni correction for the significant voxels for Route 1. The critical p -value for the Route 1 effect in the amygdala (i.e., $t(57) = 3.87$, $[x = 34, y = 2, z = -22]$, $p = .022$) survives this correction, yielding a corrected p -value of .044. We have now added this information to the relevant sentence in the **Results** section (line 393):

“For aversive value (AV), we observed a negative correlation between its difference scores and BOLD activity in the amygdala (as expected; right: $p = .022$ peak-voxel FWE-SVC; survives Bonferroni correction for two ROIs, i.e., $p_{corrected} = .044$), but unexpectedly not in the PAG (Figure 5B; for whole-brain results see Supplementary Table 4).”

8. The authors offer interpretations of the key findings separately, but this could be further developed to integrate both the aversive value and value comparison findings to offer more mechanistic view of the role of bradycardia/freezing in these neurocomputations. What are the implications if, as you speculate, the bradycardia/freezing is an epiphenomenon?

This is a very good suggestion. Though, first to briefly clarify, we speculate that bradycardia/freezing states are *not* an epiphenomenon.

We agree we should take together our findings concerning the role of freezing states in the computation of aversive value and value comparison. Here, we define freezing states as a threat-anticipatory psychophysiological state characterized by joint sympathetic and parasympathetic activation with parasympathetic dominance (causing bradycardia and immobility). We theorize that freezing states aid decision-making through upregulated processing of external sensory information such as environmental reward and threat (while accounting for the current internal bodily state; see also Roelofs & Dayan, 2022). Specifically, freezing states may primarily facilitate processing of the aversive value of potential threats (a process involving the amygdala) to ensure safety. However, the aversive value of threat should be evaluated in relation to the appetitive value of potential reward. Indeed, while it may generally be useful to avoid threat, if the potential reward is relatively high it may be worth to approach instead.

Therefore, value comparison (weighing threat versus reward) must take place. Freezing states may facilitate this value comparison process by upregulating ongoing value-based computations in the dmPFC. Facilitation of both these mechanisms (i.e., computation of aversive value and value comparison) during freezing states enables threat-coping behaviors that protect the animal from potential harm (by biasing towards avoidance) and simultaneously maintain the behavioral flexibility necessary to obtain large rewards (by biasing towards approach). Together, and more generally, our findings imply that how we weigh the potential outcomes of our actions depends on the current psychophysiological state.

We have clarified this point in the **Discussion** (line 551 - 556):

“This sensory upregulation might subsequently facilitate neural processing in downstream brain circuits involved in value assignment. This way, we theorize, freezing states may facilitate the computation of aversive value in the amygdala (increasing threat avoidance) as well as value comparison of reward and threat in the dmPFC (maintaining the behavioral flexibility to obtain large rewards). Indeed, our findings imply that how we weigh the potential outcomes of our actions depends on the current psychophysiological state. Nevertheless, causal manipulations, such as deep brain stimulation of regions critically involved in freezing states (like the periaqueductal gray (Signoret-Genest et al., 2023; Tovote et al., 2016; Walker & Carrive, 2003)), are needed to directly test this hypothesis.”

9. The authors do not report/explore relationships with questionnaire measures which seems a shame as the authors are specifically interested in links with psychopathology. The study may well be underpowered for looking at links with individual differences in measures external to the task, but these may allow for some interesting exploratory analyses.

We agree that it would be highly interesting to relate behavioral and neural findings to psychopathological symptoms. In the current study, however, the main goal of collecting anxiety and depression questionnaire scores was to characterize our sample. Particularly because this is the first application of our experimental paradigm in a neuroimaging context, such characterization is relevant to enable comparison of the current findings with those of potential future studies. Additionally, in the future, one possibility by which we may investigate these individual differences is to pool the samples of the current and future studies.

10. Relatedly, the authors report collecting pupil diameter measures, but this is not mentioned again. Were these analysed? I wouldn't necessarily expect the authors to run this additional analysis if they have reason not to, but it would be at least worth explaining in the text.

Per pre-registration we were a priori most interested in the heart rate data. Generally, and unfortunately, the pupil data collected for this study was not of great quality for several participants. Therefore, we decided to focus our efforts on analyzing the heart rate data, which is our primary physiological measure to quantify defensive freezing-like states. We have added this information to the **Methods** (line 668):

“An EyeLink 1000 eye-tracker system (SR Research, Kanata, Ontario, Canada) was used to measure pupil diameter (sampling rate of 1000 Hz). Skin conductance and pupil/eye data were not analyzed for this study because of relatively poor data quality for several participants.”

11. In general, the manuscript references previous literature appropriately, however the literature on interoception is highly relevant and not mentioned. Secondly the authors mention a need for high temporal precision MEG work, but do not mention related work in approach-avoidance paradigms (e.g., Khemka et al., 2017).

We agree that the interoception literature is of potential relevance to our manuscript. Indeed, previously we have written on the potential functional role of cardiac activity in perception and action (Skora, Livermore, & Roelofs, 2022). An exhaustive discussion of this literature in our manuscript was initially not included because we thought it is beyond the scope of the current study, given that we focus on cardiac responses as an out-read of defensive psychophysiological states rather than as the subject of active processing by the brain. Nevertheless, to better acknowledge the potential relevance of interoception to our work, we have now incorporated it more explicitly in the **Discussion** (line 546 - 549):

“We speculate that freezing states may be more than an epiphenomenon, and that they serve a role in integrating the value of external stimuli while accounting for the internal bodily state. This interpretation would be in line with findings that heart rate reductions and slow breathing rates during freezing serve sensory processing, for example by optimizing interoception of cardiac and other bodily signals (Khalsa et al., 2021), subsequently increasing neural signal-to-noise (Bagur et al., 2021; de Voogd et al., 2022; Skora et al., 2022).”

Regarding the work on high temporal precision (MEG) studies, we have re-positioned relevant references so that they more clearly link the according statement, and have now added the Khemka et al. (2017) paper (line 483):

“Future studies using neuroimaging methods with higher temporal precision, such as magnetoencephalography (MEG) (Castegnetti et al., 2020; Dumas et al., 2013; Khemka et al., 2017; Tzovara et al., 2019), might provide more insight into the role of the amygdala and associated neural circuits in value-based decision-making under threat.”

(for reviewer #2 comments please continue to next pages)

Reviewer #2 (Remarks to the Author):

Dear Editor,

I read the article written by Klaassen and colleagues titled “The neurocomputational link between defensive cardiac states and approach-avoidance arbitration under threat”.

I found, the study well conducted, and very interesting. Therefore, this manuscript can be of interest for the readers of the Communication Biology. I have to say that I do not feel sufficient expert for the modelling applied in the study and therefore I leave this judgement to other reviewers or the editor.

The study was in general clearly described and I do not have many comments, but only a few.

We thank you for your compliments, we are happy you find the manuscript interesting to read.

General Comments

- 1. While reading the abstract, I missed reading the goal as well as some details about the method.**

We agree that the abstract could more clearly describe the goal and methods of the study, however we would like to point out that we are limited to use 150 words. Nevertheless, we improved the abstract while adhering to this strict limitation:

“Avoidance, a hallmark of anxiety-related psychopathology, often comes at a cost; avoiding threat may forgo the possibility of a reward. Theories predict that optimal approach-avoidance arbitration depends on threat-induced psychophysiological states, like freezing-related bradycardia. Here we used model-based fMRI analyses to investigate whether and how bradycardia states are linked to the neurocomputational underpinnings of approach-avoidance arbitration under varying reward and threat magnitudes. We show that bradycardia states are associated with increased threat-induced avoidance and more pronounced reward-threat value comparison (i.e., a stronger tendency to approach vs. avoid when expected reward outweighs threat). An amygdala-striatal-prefrontal circuit supports approach-avoidance arbitration under threat, with specific involvement of the amygdala and dorsal anterior cingulate (dACC) in integrating reward-threat value and bradycardia states. These findings highlight the role of human freezing states in value-based decision making, relevant for optimal threat coping. They point to a specific role for amygdala/dACC in state-value integration under threat.”

2. The (post-hoc) stars in Fig. 2 are unclear to me, at least it is unclear to which comparisons they refer to.

We apologize for the confusion. The asterisks in Figure 2 represent the significance of the main effects of reward and threat on approach/avoidance choices (2A), and of the main effects of heart rate and choice on response times (2C). Thus, note that these asterisks belong to the fixed effects of the Bayesian Mixed Effects Models, not to post-hoc tests (no post-hoc tests were necessary to interpret the effects). To clarify the meaning of the asterisks, we've added the following description to the Figure 2 legend:

“Figure 2. Task effects on choice behavior and heart rate. A) Higher money and shock levels led to more approach vs avoid choices, respectively. **B)** We observed a significant average heart rate deceleration during the anticipation screen relative to the 1 s pre-trial baseline indicative of a freezing-like bradycardia state, which was numerically but not significantly more pronounced in avoid (compared to approach) trials (see main text for statistics). Trial-by-trial bradycardia (Δhr) was quantified as the average baseline-corrected heart rate across a 5-7 s time window relative to the anticipation screen onset, such that lower (relative to higher) Δhr values indicate stronger bradycardia. **C)** Stronger trial-by-trial bradycardia was associated with faster response times (for illustration purposes, individual dots reflect RT and Δhr values aggregated separately for money and shock levels, and approach and avoid choices). Moreover, we observed faster response times for avoid compared to approach decisions. Error bars indicate +/- 1 SEM. Gray-white striped shaded area in **B** reflects partial overlap between anticipation and target movement screens across different trials (i.e., movement window onset was uniformly jittered between 6 – 7 s relative to the anticipation screen onset). BPM: beats per minute; Ap.: approach; Av.: avoid; asterisks (*) indicate ‘significant’ main effects (i.e., $HDI_{95\%}$ excludes 0) of money and shocks on choice (**A**) and heart rate and choice on response times (**C**).”

3. How did the “researcher made sure participants fully understood the task”?

After the participant had read the instructions, the researcher verbally asked the participant to answer a few questions regarding the key components of the task. Specifically, the participant was asked to explain to the researcher what the reward and threat cues represented (green coins and yellow lightning bolts on the screen), how and when they could approach and avoid a target (in relation to the target's movement direction), and what the probabilities were of receiving monetary rewards/electrical shocks following approach and avoid choices. In case the participant seemed unsure or did not answer correctly, the researcher again explained the rules of the task verbally. We added this information to the **Methods** (line 606 - 607):

“Afterwards, the researcher made sure participants fully understood the task by asking them to verbally report (and, if necessary, correcting) their understanding of how to approach/avoid, and the relation between approach-avoidance and the probability of receiving one of the outcomes.”

4. The properties of the aversive US are not precisely described. The shock lasted 200 ms, was this stimulation continuous or was it a train of pulses? To me, it is unclear what is meant by “the intensity varied in 10 steps”. What was precisely the standardized calibration procedure? Were the intensities of the five shocks pre-defined?

The electrical stimulation indeed consisted of a train of pulses (250 μ s pulses at 150 Hz). We have added this information – along with a more elaborate description of the standardized calibration procedure – to the **Methods** (line 673 - 683):

“Shock durations were 200 ms (consisting of a train of 250 μ s pulses at 150 Hz), delivered at an intensity ranging from 0 – 40 V/0 – 80 mA divided in pre-defined 10 steps/levels. To determine the appropriate stimulation intensity per participant, we performed a standardized shock calibration procedure (see Klaassen et al., 2021 and de Voogd et al., 2018). Each participant received and subjectively rated exactly five shocks (always starting at intensity step 2), with the aim to converge at a shock intensity level that was experienced as uncomfortable, but not painful (i.e., rated as 4 on a scale ranging from ‘1 = not painful at all’, to ‘5 = very painful’). In brief, if a shock was rated as 3 or lower the shock intensity was increased, if a shock was rated as 5 the intensity was decreased, and if it was rated at 4 the intensity was kept the same. The average final shock intensity across shock subjects was 4.48 ± 1.72 steps (range 1-10).”

5. The two USs differed in regard to several aspects, e.g. primary vs. secondary reinforcer, the duration. Did the authors collect any subjective ratings for these two types of stimuli? Were they comparable in regards to their arousal?

There is indeed an asymmetry between the appetitive and aversive outcomes of the task. For this study specifically we did not collect subjective ratings for these two types of stimuli. However, we can address your concerns regarding this asymmetry with the results of our preceding study using the same experimental paradigm (Klaassen et al., 2021, N = 42).

First, in that study, we collected subjective attractiveness ratings for the different reward-threat prospects that participants also faced in the task (Klaassen et al., 2021: Supplementary Figure S5, pasted below for your convenience). These ratings showed that overall participants seemed to find the monetary and shock amounts similarly (un)attractive; i.e., on the subjective level there was not a clear asymmetry in the evaluation of appetitive vs. aversive outcomes. We thus suspect that the difference in the type of reinforcement between rewards and punishments (i.e., secondary vs. primary) does not substantially affect the trade-off that participants make.

Secondly, to address whether these stimuli were similar in terms of elicited amount of (physiological) arousal, in the Klaassen et al. (2021) study we tested to what extent the reward and threat magnitudes affected anticipatory skin conductance responses, a measure of sympathetic arousal (Supplementary Figure S6). Here we found that anticipatory skin conductance responses increased as a function of both the appetitive and aversive outcome magnitudes (see figure below). Together, these findings suggest that both types of stimuli evoked similar levels of parasympathetic and sympathetic arousal.

Together, based on these results we believe that – at least during choice anticipation/deliberation – participants perceive and process the potential appetitive and aversive outcomes in a similarly salient way.

6. In the discussion, there is no mention about the limitations of the study. The only one is the slow imaging method.

We agree that for some papers it is beneficial to have a separate, dedicated, limitations section in the discussion. However, in this paper we opted to discuss the study's limitations whenever they were relevant for the interpretation of the results. Below, we have highlighted two main limitation points in the discussion text. Additionally, we extended and qualified one of them more explicitly as a limitation.

For completeness - on the limited temporal resolution (line 475 - 484):

*"As such, the reduction in amygdala activity might reflect a consequence of the decision (to avoid) rather than a mechanism underlying the decision itself. Unfortunately, the **limited** temporal specificity of the fMRI signal does not allow us to separate these two possible explanations. However, our FIR time-series analysis of the BOLD response in the amygdala showed that choice effects already occurred early during the anticipation window (see Supplementary Information), hinting at early involvement of the amygdala in the approach-avoidance decision process. Future studies using neuroimaging methods with higher temporal precision, such as magnetoencephalography (MEG) (Castegnetti et al., 2020; Dumas et al., 2013; Khemka et al., 2017; Tzovara et al., 2019), might provide more insight into the role of the amygdala and associated neural circuits in value-based decision-making under threat."*

On the interpretational limitation regarding inference of causality (line 533 - 535), now explicitly mentioned as a limitation:

*"One open question pertains to the causal chain of effects regarding the link between freezing states and value-based computations underlying approach-avoidance. **Indeed, the interpretation of our findings is limited by the correlational nature of our experimental design.** For example, freezing states might not causally contribute to changes in value computations but rather be an epiphenomenon of task engagement."*

Minor comments

7. I am not sure the term "fearful avoidance" is the most appropriate. Avoidance can be elicited by a threat or by the feeling of fear, but it cannot be fearful per sé.

We understand your point and to avoid confusion we have removed the term 'fearful' from the manuscript (two instances):

In the **Abstract** (line 33):

"~~Fearful~~ Avoidance is a hallmark of anxiety disorders and often comes at a cost."

In the **Introduction** (line 50):

"~~Fearful-Threat~~ avoidance often comes at a cost, particularly in approach-avoidance conflict where avoidance may reduce the probability of aversive outcomes but also of obtaining potential rewards."

8. Specify the size of the electrodes used.

We added this information now to the **Methods** (line 671):

“Electrical shocks were delivered through two 10 mm Ag/AgCl electrodes attached to the distal phalanges of the third and fourth fingers of the right hand using a MAXTENS 2000 (Bio-Protech) machine.”

9. Did the authors apply any filter for the physiological responses?

Yes we applied some filtering to preprocess the physiological data. MR artifacts were identified as the modulation of the physiological signal relative to each TR and subsequently removed using deconvolution. The cleaned heart rate signals were high-pass filtered at 0.01 Hz, and low-pass filtered at 2.5 Hz. We added this information to the **Methods** (line 699 - 702):

“Raw pulse and respiratory data were preprocessed using in-house software for removal of radio frequency artifacts and interactive visual artifact correction and peak detection (<https://github.com/can-lab/brainampconverter>, <https://github.com/can-lab/hera>). In general, MR artifacts were identified as the modulation of the signal relative to each TR and subsequently removed using deconvolution. The cleaned heart rate signals were high-pass filtered at 0.01 Hz, and low-pass filtered at 2.5 Hz.”

10. Fig.3 and Fig 5. depict correlations. Close to the scale, I found the abbreviation “uncorr.” misleading. It refers to uncorrected, but because of the correlations depicted in the picture I first thought that it means uncorrelated.

We understand the confusion and therefore now removed the term ‘uncorr.’ from the Figures, and specified the image thresholding in the Figure captions:

“Figure 3. Neural correlates of reward-threat and approach-avoidance anticipation. A) We observed positive correlations between BOLD and money levels in the ventral striatum (vStr), and between BOLD and shock levels in the supplemental motor area/dorsal anterior cingulate (SMA/dACC) and anterior insula. **B)** We observed higher BOLD activity in the ventral striatum, amygdala, and ventromedial prefrontal cortex (vmPFC) for approach compared to avoid choices. *Images are thresholded at $p < .001$ whole-brain uncorrected for display purposes.* All labelled areas are significant at $p < .05$ FWE-corrected.”

“Figure 5. Neural circuits underlying the link between bradycardia states and value-based computations. A) For the base model, we find positive correlations between model-extracted decision values (DVs, indicating the value of approaching vs. avoiding) and BOLD activity in the ventral striatum (vStr) and ventromedial prefrontal cortex (vmPFC), delineating a similar neural circuit as was observed from the task-based analysis (see Fig. 3). **B)** DVs estimated from the aversive value (AV) model were, relative to the base model, negatively correlated with BOLD activity in the amygdala (top). A follow-up analysis revealed that this negative relationship was specific to conditions in which AV predicted more avoidance, compared to the base model (i.e., $DV_{diff_{AV}} > 0$; bottom). **C)** DVs estimated from the value comparison (VC) model were, relative to the base model, positively correlated to BOLD in a large whole brain-significant cluster in the dorsomedial prefrontal cortex (dmPFC), spanning the SMA and dACC (top). Follow-up analysis revealed that this relationship was specific to conditions in which VC predicted more

approach, compared to the base model (i.e., $Dvdiff_{VC} < 0$; bottom). $Dvdiff_{AV/VC}$: trial-wise difference scores of the DVs of AV/VC models relative to the base model. Positive/negative $Dvdiff_{AV/VC}$ values indicate stronger predicted probability to avoid/approach by AV/VC models compared to the base model. Error bars indicate ± 1 SEM. **Images are thresholded at $p < .005$ whole-brain uncorrected for display purposes.** All labelled areas are significant at $p < .05$ FWE-corrected.”

REFERENCES

- Avery, S. N., Clauss, J. A., Winder, D. G., Woodward, N., Heckers, S., & Blackford, J. U. (2014). BNST neurocircuitry in humans. *NeuroImage*, *91*, 311–323.
<https://doi.org/10.1016/j.neuroimage.2014.01.017>
- Bagur, S., Lefort, J. M., Lacroix, M. M., de Lavilléon, G., Herry, C., Chouvaeff, M., Billand, C., Geoffroy, H., & Benchenane, K. (2021). Breathing-driven prefrontal oscillations regulate maintenance of conditioned-fear evoked freezing independently of initiation. *Nature Communications*, *12*(1), 1–15. <https://doi.org/10.1038/s41467-021-22798-6>
- Castegnetti, G., Tzovara, A., Khemka, S., Melinščak, F., Barnes, G. R., Dolan, R. J., & Bach, D. R. (2020). Representation of probabilistic outcomes during risky decision-making. *Nature Communications*, *11*(1), 1–11. <https://doi.org/10.1038/s41467-020-16202-y>
- Christopoulos, G. I., Tobler, P. N., Bossaerts, P., Dolan, R. J., & Schultz, W. (2009). Neural correlates of value, risk, and risk aversion contributing to decision making under risk. *The Journal of Neuroscience*, *29*(40), 12574–12583. <https://doi.org/10.1523/JNEUROSCI.2614-09.2009>
- de Voogd, L. D., Hagenberg, E., Zhou, Y. J., de Lange, F. P., & Roelofs, K. (2022). Acute threat enhances perceptual sensitivity without affecting the decision criterion. *Scientific Reports*, *12*(1), 1–11. <https://doi.org/10.1038/s41598-022-11664-0>
- Dumas, T., Dubal, S., Attal, Y., Chupin, M., Jouvent, R., Morel, S., & George, N. (2013). MEG Evidence for Dynamic Amygdala Modulations by Gaze and Facial Emotions. *PLoS ONE*, *8*(9), e74145. <https://doi.org/10.1371/journal.pone.0074145>

- Gladwin, T. E., Hashemi, M. M., van Ast, V., & Roelofs, K. (2016). Ready and waiting: Freezing as active action preparation under threat. *Neuroscience Letters*, *619*, 182–188.
<https://doi.org/10.1016/j.neulet.2016.03.027>
- Hespanhol, L., Vallio, C. S., Costa, L. M., & Saragiotto, B. T. (2019). Understanding and interpreting confidence and credible interval around effect estimates. *Brazilian Journal of Physical Therapy*, *23*(4), 290–301. <https://doi.org/10.1016/j.bjpt.2018.12.006>
- Hiser, J., & Koenigs, M. (2018). The Multifaceted Role of the Ventromedial Prefrontal Cortex in Emotion, Decision Making, Social Cognition, and Psychopathology. *Biological Psychiatry*, *83*(8), 638–647.
<https://doi.org/10.1016/j.biopsych.2017.10.030>
- Hulsman, A. M., Terburg, D., Roelofs, K., & Klumpers, F. (2021). Roles of the bed nucleus of the stria terminalis and amygdala in fear reactions. In D. F. Swaab, F. Kreier, P. J. Lucassen, A. Salehi, & R. M. Buijs (Eds.), *Handbook of Clinical Neurology* (Vol. 179, pp. 419–432). Elsevier.
<https://doi.org/10.1016/B978-0-12-819975-6.00027-3>
- Khalsa, S. S., Gar, S. N., Paulus, M. P., & Koch, C. (2021). Computational Models of Interoception and Body Regulation. *Trends in Neurosciences*, *44*(1), 63–76. <https://doi.org/10.1016/j.tins.2020.09.012>
- Khemka, S., Barnes, G., Dolan, R. J., & Bach, D. R. (2017). Dissecting the Function of Hippocampal Oscillations in a Human Anxiety Model. *The Journal of Neuroscience*, *37*(29), 6869–6876.
<https://doi.org/10.1523/JNEUROSCI.1834-16.2017>
- Kirlic, N., Young, J., & Aupperle, R. L. (2017). Animal to human translational paradigms relevant for approach avoidance conflict decision making. *Behaviour Research and Therapy*, *96*, 14–29.
<https://doi.org/10.1016/j.brat.2017.04.010>
- Klaassen, F. H., Held, L., Figner, B., O'Reilly, J. X., Klumpers, F., de Voogd, L. D., & Roelofs, K. (2021). Defensive freezing and its relation to approach-avoidance decision-making under threat. *Scientific Reports*, *11*(12030). <https://doi.org/10.1038/s41598-021-90968-z>

- Klumpers, F., Kroes, M. C. W., Baas, J. M. P., & Fernández, G. (2017). How human amygdala and bed nucleus of the stria terminalis may drive distinct defensive responses. *The Journal of Neuroscience*, *37*(40), 9645–9656. <https://doi.org/10.1523/JNEUROSCI.3830-16.2017>
- Kolling, N., Wittmann, M., & Rushworth, M. F. S. (2014). Multiple neural mechanisms of decision making and their competition under changing risk pressure. *Neuron*, *81*(5), 1190–1202. <https://doi.org/10.1016/j.neuron.2014.01.033>
- Livermore, J. J. A., Klaassen, F. H., Bramson, B., Hulsman, A. M., Meijer, S. W., Held, L., Klumpers, F., de Voogd, L. D., & Roelofs, K. (2021). Approach-Avoidance Decisions Under Threat: The Role of Autonomic Psychophysiological States. *Frontiers in Neuroscience*, *15*(March), 1–12. <https://doi.org/10.3389/fnins.2021.621517>
- Lojowska, M., Gladwin, T. E., Hermans, E. J., & Roelofs, K. (2015). Freezing promotes perception of coarse visual features. *Journal of Experimental Psychology: General*, *144*(6), 1080–1088. <https://doi.org/10.1037/xge0000117>
- Löw, A., Weymar, M., & Hamm, A. O. (2015). When Threat Is Near, Get Out of Here: Dynamics of Defensive Behavior During Freezing and Active Avoidance. *Psychological Science*, *26*(11), 1706–1716. <https://doi.org/10.1177/0956797615597332>
- Mobbs, D., Marchant, J. L., Hassabis, D., Seymour, B., Tan, G., Gray, M., Petrovic, P., Dolan, R. J., & Frith, C. D. (2009). From Threat to Fear: The Neural Organization of Defensive Fear Systems in Humans. *The Journal of Neuroscience*, *29*(39), 12236–12243. <https://doi.org/10.1523/JNEUROSCI.2378-09.2009>
- Mobbs, D., Petrovic, P., Marchant, J. L., Hassabis, D., Weiskopf, N., Seymour, B., Dolan, R. J., & Frith, C. D. (2007). When Fear Is Near: Threat Imminence Elicits Prefrontal-Periaqueductal Gray Shifts in Humans. *Science*, *317*(5841), 1079–1083. <https://doi.org/10.1126/science.1144298>

- Neubert, F., Mars, R. B., Sallet, J., & Rushworth, M. F. S. (2015). Connectivity reveals relationship of brain areas for reward-guided learning and decision making in human and monkey frontal cortex. *Proceedings of the National Academy of Sciences*, 1–10.
<https://doi.org/10.1073/pnas.1410767112>
- Piray, P., Ouden, H. E. M. D., Schaaf, M. E. V. D., Toni, I., & Cools, R. (2017). Dopaminergic Modulation of the Functional Ventrodorsal Architecture of the Human Striatum. *Cerebral Cortex*, 27(October 2015), 485–495. <https://doi.org/10.1093/cercor/bhv243>
- Quandt, J., Figner, B., Holland, R. W., & Veling, H. (2021). Confidence in evaluations and value-based decisions reflects variation in experienced values. *Journal of Experimental Psychology: General*, 151(4), 820–836. <https://doi.org/10.1037/xge0001102>
- Roelofs, K. (2017). Freeze for action: Neurobiological mechanisms in animal and human freezing. *Philosophical Transactions of the Royal Society B: Biological Sciences*, 372(1718).
<https://doi.org/10.1098/rstb.2016.0206>
- Rolls, E. T., Huang, C. C., Lin, C. P., Feng, J., & Joliot, M. (2020). Automated anatomical labelling atlas 3. *NeuroImage*, 206(September 2019), 116189.
<https://doi.org/10.1016/j.neuroimage.2019.116189>
- Rolls, E. T., McCabe, C., & Redoute, J. (2008). Expected Value, Reward Outcome, and Temporal Difference Error Representations in a Probabilistic Decision Task. *Cerebral Cortex*, 18(3), 652–663.
<https://doi.org/10.1093/cercor/bhm097>
- Roy, M., Shohamy, D., Daw, N., Jepma, M., Wimmer, G. E., & Wager, T. D. (2014). Representation of aversive prediction errors in the human periaqueductal gray. *Nature Neuroscience*, 17(11), 1607–1612. <https://doi.org/10.1038/nn.3832>

- Signoret-Genest, J., Schukraft, N., Reis, S. L., Segebarth, D., Deisseroth, K., & Tovote, P. (2023). Integrated cardio-behavioral responses to threat define defensive states. *Nature Neuroscience*.
<https://doi.org/10.1038/s41593-022-01252-w>
- Skora, L. I., Livermore, J. J. A., & Roelofs, K. (2022). The functional role of cardiac activity in perception and action. *Neuroscience and Biobehavioral Reviews*, *137*(October 2021), 104655.
<https://doi.org/10.1016/j.neubiorev.2022.104655>
- Tovote, P., Esposito, M. S., Botta, P., Chaudun, F., Fadok, J. P., Markovic, M., Wolff, S. B. E., Ramakrishnan, C., Fenno, L., Deisseroth, K., Herry, C., Arber, S., & Lüthi, A. (2016). Midbrain circuits for defensive behaviour. *Nature*, *534*(7606), 206–212. <https://doi.org/10.1038/nature17996>
- Tzovara, A., Meyer, S. S., Bonaiuto, J. J., Abivardi, A., Dolan, R. J., Barnes, G. R., & Bach, D. R. (2019). High-precision magnetoencephalography for reconstructing amygdalar and hippocampal oscillations during prediction of safety and threat. *Human Brain Mapping*, *40*(14), 4114–4129.
<https://doi.org/10.1002/hbm.24689>
- van Duijvenvoorde, A. C. K., Huizenga, H. M., Somerville, L. H., Delgado, M. R., Powers, A., Weeda, W. D., Casey, B. J., Weber, E. U., & Figner, B. (2015). Neural Correlates of Expected Risks and Returns in Risky Choice across Development. *The Journal of Neuroscience*, *35*(4), 1549–1560.
<https://doi.org/10.1523/JNEUROSCI.1924-14.2015>
- Walker, P., & Carrive, P. (2003). Role of ventrolateral periaqueductal gray neurons in the behavioral and cardiovascular responses to contextual conditioned fear and poststress recovery. *Neuroscience*, *116*(3), 897–912. [https://doi.org/10.1016/S0306-4522\(02\)00744-3](https://doi.org/10.1016/S0306-4522(02)00744-3)
- Wendt, J., Löw, A., Weymar, M., Lotze, M., & Hamm, A. O. (2017). Active avoidance and attentive freezing in the face of approaching threat. *NeuroImage*, *158*(December 2016), 196–204.
<https://doi.org/10.1016/j.neuroimage.2017.06.054>

REVIEWERS' COMMENTS:

Reviewer #1 (Remarks to the Author):

The authors have addressed my comments very thoroughly and the manuscript reads much more clearly as a result. This is a very interesting study, and I have nothing further to suggest.

Reviewer #2 (Remarks to the Author):

I thank the authors for having clarified all my concerns. I do not have any other comments.

REVIEWERS' COMMENTS:

Reviewer #1 (Remarks to the Author):

The authors have addressed my comments very thoroughly and the manuscript reads much more clearly as a result. This is a very interesting study, and I have nothing further to suggest.

Reviewer #2 (Remarks to the Author):

I thank the authors for having clarified all my concerns. I do not have any other comments.

Author Response:

We thank both reviewers for their useful comments and feedback. It helped us to improve the quality of our paper.